# X-VLA: Soft-Prompted Transformer as Scalable Cross-Embodiment Vision-Language-Action Model

**Jinliang Zheng**[1,2*], **Jianxiong Li**[1,*], **Zhihao Wang**[1,3], **Dongxiu Liu**[1], **Xirui Kang**[1],
**Yuchun Feng**[1], **Yinan Zheng**[1], **Jiayin Zou**[1], **Yilun Chen**[2], **Jia Zeng**[2], **Ya-Qin Zhang**[1],
**Jiangmiao Pang**[2], **Jingjing Liu**[1], **Tai Wang**[2†], **Xianyuan Zhan**[1†]

[1] Tsinghua University   [2] Shanghai AI Laboratory   [3] Peking University
[*] Equal Contribution   [†]Corresponding Author
wangtai@pjlab.org.cn & zhanxianyuan@air.tsinghua.edu.cn
**Project website**: https://thu-air-dream.github.io/X-VLA/

## Abstract

Successful generalist Vision-Language-Action (VLA) models rely on effective training across diverse robotic platforms with large-scale, cross-embodiment, heterogeneous datasets. To facilitate and leverage the heterogeneity in rich, diverse robotic data sources, we propose a novel *Soft Prompt* approach with minimally added parameters, by infusing prompt learning concepts into cross-embodiment robot learning and introducing separate sets of learnable embeddings for each distinct data source. These embeddings serve as embodiment-specific prompts, which in unity empower VLA models with effective exploitation of varying cross-embodiment features. Our new *X-VLA*, a neat flow-matching-based VLA architecture, relies exclusively on soft-prompted standard Transformer encoders with an enhanced encoding pipeline, enjoying both scalability and simplicity. Evaluated across 6 simulation environments as well as 3 real-world robotics platforms, our 0.9B instantiation-*X-VLA-0.9B* simultaneously achieves state-of-the-art performance over a sweep of benchmark suites, demonstrating superior results on a wide range of capabilities, from flexible dexterity to quick adaptation across embodiments, environments, and tasks.

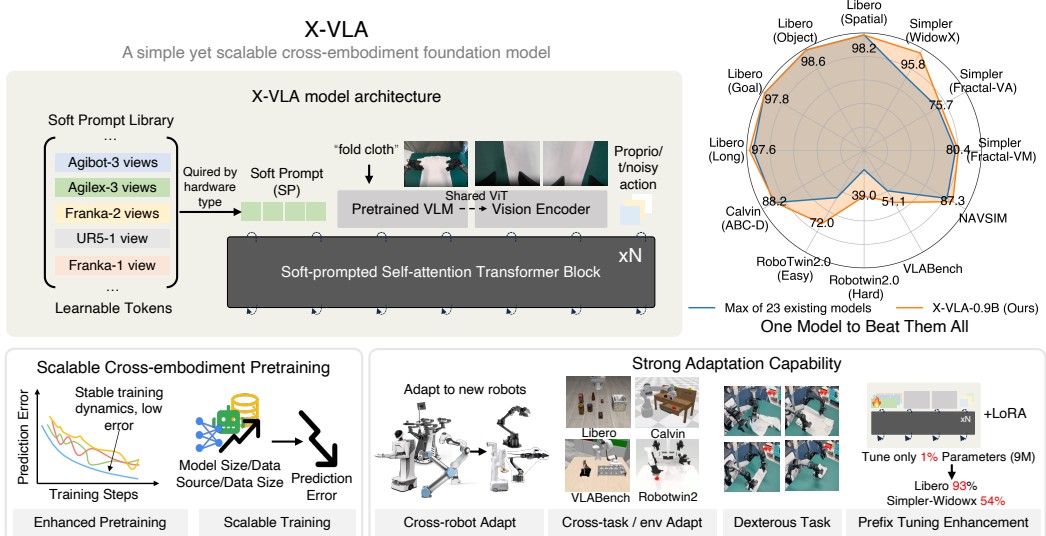

Figure 1: *X-VLA* employs distinctive learnable embeddings, referred to as *soft prompt*, to effectively address the heterogeneity present in cross-embodiment datasets. Combined with stacked self-attention transformer blocks and an enhanced encoding pipeline for multimodal inputs, this design offers a scalable framework for integrating diverse pretraining datasets and adapting to domain-specific applications. Evaluated across 6 simulation benchmark including five manipulation benchmarks and one autonomous driving benchmark, as well as 3 real-world robots, *X-VLA* achieves SOTA performance over most benchmark suites and real-world robotic tasks.

# 1 INTRODUCTION

It has long been a central ambition in the robotics community (Brohan et al., 2023; 2022) to build autonomous agents that are capable of: (1) flexibly following arbitrary human instructions (Wang et al., 2025b), and (2) dexterously operating across diverse environments as well as disparate embodiments (Black et al., 2025a). In light of recent success of Large Language Models (LLMs) (Achiam et al., 2023; Bai et al., 2023a; Touvron et al., 2023) and Vision-Language Models (VLMs) (Li et al., 2024a; Bai et al., 2023b), one promising direction is to extend these advanced architectures to robotics through the incorporation of precise action modalities, thereby giving rise to Vision-Language-Action (VLA) models (Kim et al.). The inherent expectation is that such large VLA models can marry out-of-the-box generalization with robust manipulation capabilities, from simple pick-and-place operations to complex dexterous tasks (Black et al., 2025b; Team et al., 2025; NVIDIA et al., 2025).

The success of VLA models, particularly their ability to rapidly adapt to out-of-distribution (OOD) domains, hinges on pretraining with large and diverse robotics datasets that encompass a broad spectrum of robotic system architectures and a wide range of task scenarios (O'Neill et al., 2024; Lin et al., 2025). A key challenge here is that VLA models face substantial heterogeneity from hardware configurations to data collection strategies (Wang et al., 2024b). Such heterogeneity manifests not only in embodiment-specific action spaces (Liu et al., 2025b), but also in setup variations such as camera settings, visual domains, and task distributions (Doshi et al., 2024b; Shi et al., 2025b; Zhen et al., 2024). These various dimensions of diversity induce severe distributional shifts as well as significant semantic misalignments across embodiments, confusing the model and ultimately leading to unsatisfactory pretraining and adaptation performance (Zheng et al., 2025; Liu et al., 2025b). Existing VLA methods primarily assign distinct action decoder heads to accommodate embodiment-specific action spaces as their main focus (Black et al., 2025a; NVIDIA et al., 2025), with other critical sources of heterogeneity ineluctably overlooked. Reconciliation among these disparate configurations, however, is crucial for proprioceptive-aware reasoning and for distilling shared knowledge from heterogeneous, mixed-domain datasets, which persistently remains an unsolved problem due to: (1) the inconsistency across hardware platforms, (2) the absence of standardized data collection protocols, and (3) the inherent domain shifts that arise across embodiment and environment barriers (O'Neill et al., 2024).

We demonstrate that these obstacles can be effectively overcome with minimal human effort involved, by allowing VLA models to learn domain-specific hardware configurations through a simple *Soft Prompt* mechanism (Lester et al., 2021). Inspired by insights from meta-learning and multi-task learning, we recast diverse hardware configurations and data types in robotics domain into the mold of task-specific features, which can then be effectively captured through prompt-learning techniques (Wang et al., 2023; Liu et al., 2023c; Khattak et al., 2023; Liu et al., 2023b). Specifically, to model the varying dimensions of heterogeneity as aforementioned, we assign a set of learnable embeddings to each data source as *Soft Prompts*. These embeddings provide heterogeneity-aware guidance for structuring the VLA representation space from early stages of feature fusion, which endows the VLA model with an enhanced capacity to exploit and consolidate cross-embodiment variations, improving generalization across different hardware and task configurations.

Formally, we introduce *X-VLA*, a generalist flow-matching–based VLA framework built upon a Soft-prompted Transformer, designed to operate seamlessly across heterogeneous platforms. Through *Soft Prompts*, *X-VLA* can be guided by explicitly-learned individual hardware setups to accommodate various structures of system and data. With a versatile architecture well-equipped for simultaneously encoding multi-view images, language prompts and proprioceptive features, *X-VLA* allows scalable VLA training, by simply stacking standard Transformer encoders (Vaswani et al., 2017) for multimodal feature fusion and precise action generation.

Extensive experiments demonstrate that *Soft Prompts* outperform other state-of-the-art (SOTA) methods in handling various heterogeneity dimensions. The *X-VLA* architecture exhibits a stable learning process and superior asymptotic performance, offering favorable scaling capabilities towards larger model size and mixed-robot datasets.

We implement *X-VLA-0.9B*, a 0.9B instantiation of *X-VLA*, trained with a carefully designed data processing and learning recipe. The overall training pipeline consists of two phases: **Phase I: Pretraining.** We pretrain *X-VLA-0.9B* on a curated heterogeneous data mixture comprising 290K

episodes (Khazatsky et al., 2024; Wu et al., 2024; Bu et al., 2025), spanning seven platforms across five types of robotic arms, ranging from single-arm to bi-manual setups. By leveraging soft prompts to absorb embodiment-specific variations, the model learns an embodiment-agnostic generalist policy. **Phase II: Domain adaptation.** *X-VLA-0.9B* is adapted into a deployable policy for a target domain. A new set of soft prompts is introduced and optimized to encode the hardware configuration of the novel domain, while the pretrained backbone remains frozen. With these prompts in place, the policy is then effectively specialized to the new embodiment through finetuning.

We conduct extensive experiments to evaluate the adaptation capabilities of the proposed *X-VLA-0.9B* across diverse embodiments, environments, and tasks. Remarkably, X-VLA-0.9B achieves new state-of-the-art results on six simulation benchmarks, including five robotics benchmarks and one autonomous driving benchmark, as well as on three real-world robot platforms. Furthermore, with only 1,200 demonstrations, the model excels at dexterous cloth-folding manipulation in the real world, achieving an average throughput of folding a cloth in under two minutes—comparable to closed-source models with substantially more parameters trained on larger datasets. In addition, we demonstrate that Phase II adaptation can be efficiently realized through parameter-efficient finetuning (Hu et al., 2022) at minimal training cost. Specifically, with the aid of previously learned prompts, X-VLA-0.9B achieves a 93% success rate on LIBERO and 54% on Simpler-WidowX by tuning only 1% of the model parameters (9M). These results are comparable to $\pi_0$ (Black et al., 2025a), despite requiring 300× fewer parameters (9M vs. 3B).

## 2 PRELIMINARY

**VLA models.** VLA models are a class of models that unify multi-modal understanding and action generation for robotic control (Black et al., 2025a; NVIDIA et al., 2025). Typically, VLA models are initialized from VLMs pretrained on large-scale image-text corpora, and then finetuned on robotics dataset containing expert trajectories: $\mathcal{D} = \{\tau_j\}_{j=1}^M$, $\tau_j = \{(o_n, a_n)\}_{n=1}^{N_j}$, where $o_n$ denotes multi-modal observation at step $n$ (e.g., visual input, language instruction, proprioceptive states), and $a_n$ is its corresponding expert action. The training objective is typically framed as behavior cloning, where the policy $\pi_\theta(o_n)$ parameterized by $\theta$ is optimized to predict the demonstrated action chunk $A_n := [a_n, a_{n+1}, ..., a_{n+T}]$ where $T$ denotes the chunk size, by minimizing a suitable supervised loss $\ell(\cdot)$ as: $\mathcal{L}_{\mathrm{BC}}(\theta) = \mathbb{E}_{(o_n, A_n) \sim \mathcal{D}}\big[\ell\big(\pi_\theta(o_n), A_n\big)\big]$.

**Flow-matching policy.** Instead of directly predicting the expert action chunk $A$ from observation $o$, flow-matching policies commonly learns a velocity field (Lipman et al., 2023) that transports a noise sample to the target action chunk. For instance, one can generate an action $A$ by starting from an Gaussian noise $A^0 \sim \mathcal{N}(0, I)$ and iteratively refining it through a velocity field $v_\theta(A^t, o, t)$ parameterized by a neural network using ODE solvers such as Euler-Maruyama method: $A^{t+\Delta t} = A^t + v_\theta(A^t, o, t)\Delta t$. Here, $t \in [0, 1]$ is a continuous time variable. To train the velocity field, we use the OT (optimal transport) path (Lipman et al., 2024; 2023), which aligns the velocity with the linear interpolated path between noise and expert data:

$$\mathcal{L}_{\mathrm{BC}}^{\mathrm{FM}}(\theta) = \mathbb{E}_{t \sim \mathcal{U}(0,1),\, (o,A) \sim \mathcal{D}}\Big[\big\|v_\theta(A^t, o, t) - (A - A^0)\big\|^2\Big],$$

where $A^t = (1 - t)A^0 + tA$, $\mathcal{U}$ is uniform distribution. By minimizing $\mathcal{L}_{\mathrm{BC}}^{\mathrm{FM}}$, the policy learns to progressively transport random noise toward expert chunks conditioned on observations.

**Heterogeneity in cross-embodiment training.** Training on mixed data recipes composed of $H$ heterogeneous datasets, $\mathcal{D}^H = \{\mathcal{D}_i\}_{i=1}^H$, is essential for developing generalist VLA models (Doshi et al., 2024a; O'Neill et al., 2024). Such training enables the aggregation of knowledge across diverse robots, facilitating fast cross-embodiment transfer and out-of-the-box deployment on unseen platforms. Each dataset, $\mathcal{D}_i$, is collected under a specific hardware configuration, $h_i \in \mathcal{H}$, where $\mathcal{H}$ represents the space of possible hardware setups, such as arm kinetics, control interfaces, camera setups, and deployment scenarios. These introduce significant heterogeneity, not only in low-level action signals and distributions, but also in high-level visual understanding, which can result in poor pretraining and adaptation if not effectively addressed (Wang et al., 2024b; Zheng et al., 2025).

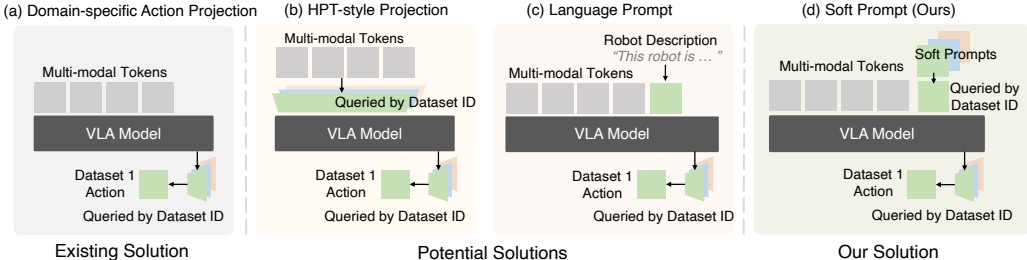

Figure 2: Comparison among four methods in handling heterogeneity in cross-embodiment training.

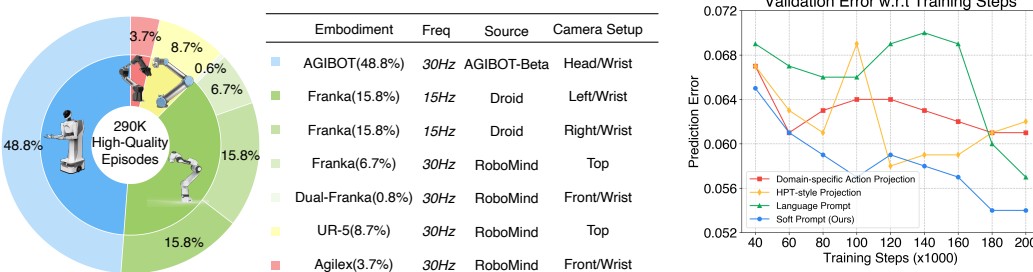

| Embodiment | Freq | Source | Camera Setup |
|---|---|---|---|
| AGIBOT(48.8%) | 30Hz | AGIBOT-Beta | Head/Wrist |
| Franka(15.8%) | 15Hz | Droid | Left/Wrist |
| Franka(15.8%) | 15Hz | Droid | Right/Wrist |
| Franka(6.7%) | 30Hz | RoboMind | Top |
| Dual-Franka(0.8%) | 30Hz | RoboMind | Front/Wrist |
| UR-5(8.7%) | 30Hz | RoboMind | Top |
| Agilex(3.7%) | 30Hz | RoboMind | Front/Wrist |

Figure 3: The recipe for mixed data used in pretraining experiments.

Figure 4: Training curves for various methods of handling heterogeneity.

# 3 HETEROGENEOUS SOFT PROMPT LEARNING

To address heterogeneity, we conduct a comprehensive empirical study to explore potential design choices, as shown in Fig. 2. We follow Reuss et al. (2025); NVIDIA et al. (2025) to establish a standard dual-system architecture as our starting point, which leverages VLMs for multimodal perception and a DiT-style decoder for action generation. In Fig. 3, we construct a heterogeneous data mixture from recent high-quality sources, including AGIBOT-beta (Bu et al., 2025), RoboMind (Wu et al., 2024), and Droid (Khazatsky et al., 2024). This dataset spans seven hardware setups across five robots, ranging from single-arm to bi-manual setups, providing sufficient scale and diversity necessary for generalist policy training. We evaluate all methods using a fully aligned training recipe to ensure a fair comparison. See Appendix I for more training details.

**(a) Domain-specific action projection.** This strategy addresses heterogeneity by assigning separate projection heads at the model output to map action tokens into embodiment-specific action spaces. While this approach is widely used in prior embodied foundation models (Black et al., 2025a; NVIDIA et al., 2025; Team et al., 2025; Zheng et al., 2025; Liu et al., 2025b), its effect is limited to the final action generation stage. Consequently, it fails to encourage embodiment-aware reasoning earlier in the pipeline and overlooks other critical sources of heterogeneity, such as variations in different camera setups and task distributions. To circumvent these limitations, we identify three representative strategies that improve pretraining stability on heterogeneous datasets, as summarized in Fig. 2. We analyze these strategies in the following discussion, with additional experimental attempts reported in Appendix E.

**(b) HPT-style projection.** Inspired by Wang et al. (2024b), this approach aims to mitigate domain discrepancies in observation inputs and promote generalist reasoning by mapping observations from distinct domains into a shared representation space. Specifically, domain-specific projection layers are also applied on top of multi-modal inputs to align them before being fed into the backbone.

**(c) Language prompts.** Another strategy leverages the language reasoning capabilities of pretrained VLMs. In this case, natural language descriptions of hardware configurations $h_i$ are provided as additional inputs, enabling the model to attend to embodiment-specific variations through textual descriptions explicitly. The language template used are summarized in Tab. 12.

**(d) Soft prompts.** Finally, we investigate a soft-prompt method that follows the meta-learning and multi-task learning philosophy (Finn et al., 2017; Liu et al., 2023c) by introducing domain-specific learnable parameters $P^H = \{p_i\}_{i=1}^{H}$ to absorb heterogeneity across data sources. $p_i$ is expected to encode the underlying hardware configuration: $p_i \approx \Phi(h_i)$, where $\Phi : \mathcal{H} \to \mathbb{R}^k$ denotes a

latent mapping from hardware configurations to the prompt space. Notably, $\Phi$ is not predefined as in language prompts (c) but is randomly initialized and then implicitly optimized through end-to-end training. These soft prompts are injected into the model at the early stage of action generation, automatically guiding the backbone toward embodiment-aware learning.

While (b) and (c) are conceptually appealing, they suffer from notable limitations. *HPT-style projection* introduces different projection layers in the middle of the observation processing, which frequently alter feature distributions and are prone to corrupting pretrained VLM representations, often resulting in unstable training dynamics. *Language prompts*, on the other hand, rely on carefully scripted textual descriptions of hardware configurations, which greatly hinder adaptability and scalability in practice. In contrast, soft prompts offer a flexible and scalable solution for encoding domain-specific hardware configurations. They marry the advantages of both (b) and (c), integrating smoothly with the backbone while preserving pretrained representations and eliminating the need for handcrafted annotations. The empirical results in Fig. 4 confirm that *Soft Prompts* consistently achieve much more robust and stable training across heterogeneous datasets. We further discuss additional unsuccessful attempts to address heterogeneity in Appendix E; while these approaches did not succeed, we report them to inform and inspire future research.

## 4 X-VLA: SOFT-PROMPTED TRANSFORMER ENHANCED VLA MODEL

Building on *Soft Prompts*, we introduce *X-VLA*, a neat VLA architecture designed for stable pretraining on heterogeneous datasets and efficient adaptation on new domains. In this section, we first present the overall architectural design, followed by several key techniques for large-scale pretraining. The complete ablation path is provided in Tab. 1, which highlights the contributions of the components introduced in this section.

### 4.1 ARCHITECTURE

We aim to build a streamlined encoding pipeline for complex multimodal inputs. Beyond *Soft Prompts*, *X-VLA* handles *(1)* high-dimensional inputs (multi-view visuals and languages), and *(2)* low-dimensional states (proprioception and action tokens). Due to substantial discrepancies in both semantics and dimensionality across these modalities, we employ dedicated encoding strategies to align them effectively, after which vanilla transformer stacks suffice for scalable policy learning. Below, we detail the encoding pipeline with the complete architecture and additional design explorations are provided in Fig. 5 and D.

(1) **High-dimensional observation stream**. High-dimensional inputs include multi-view images $\text{Img} = \{\text{img}_i\}$, together with languages $L$ specifying task objectives. Unlike most prior approaches (Black et al., 2025a) that directly feed all views and instructions into VLMs, we disentangle the streams by assigning distinct encoders. A pretrained VLM encoder (Florence-Large (Xiao et al., 2024)

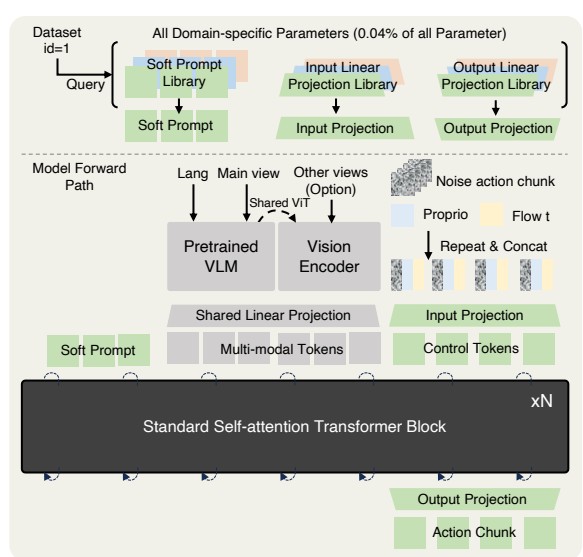

Figure 5: Illustration of our model architecture. Most parameters are shared across embodiments, with only the soft prompt and the action-related input/output projections remaining domain-specific (0.04% of total).

in *X-VLA*) is used for the main vision-language stream (fixed-view and instruction), while auxiliary views such as wrist-views are processed with a shared vision backbone. This design alleviates the semantic gap between generic vision-language reasoning and embodied reasoning: fixed-camera views provide stable, informative context for high-level task reasoning; whereas wrist-camera

inputs, though noisy and fast-changing, offer critical cues for fine-grained manipulation and are thus encoded separately from the language stream.

(2) **Low-dimensional proprioceptive–action stream**. The proprioceptive states $R^t$, such as joint positions and end-effector poses, provide embodiment-specific grounding for reasoning and control. The action-related tokens $A^t$ consist of noisy action samples used for flow-matching generation. Since both $R^t$ and $A^t$ are compact vectors with closely related physical semantics, we concatenate them along with their corresponding time embeddings $t$ within the flow-matching pipeline. The fused representation is then projected into a high-dimensional feature space through a lightweight linear layer, enabling early fusion with other modalities and ensuring robust proprioceptive–temporal grounding during both training and inference.

(3) **Soft prompts**. The well-encoded multimodal inputs are concatenated with soft prompts, which are typically a sequence of N learnable tokens. Each domain is associated with its own set of soft prompts and is selected according to the dataset id. For instance, in our pretraining setup as shown in Fig. 3, we maintain 7 distinct sets of learnable tokens corresponding to the 7 datasets. The concatenated token sequence is then passed into a standard Transformer encoder, which performs cross-modal reasoning and generates the corresponding action outputs.

## 4.2 CUSTOMIZED TRAINING RECIPE

To fully incentivize the potential of *X-VLA*, we introduce a carefully designed learning engineering to enhance both stability and effectiveness for *X-VLA* training. We provide an overview of our training recipe and outline several key techniques that are crucial for the stable and efficient training.

### 4.2.1 PRETRAINING AND FINETUNING PIPELINE

For pretraining, the backbone $\pi_\theta$ and the soft prompts $P^H$ are jointly optimized under the flow-matching objective $\mathcal{L}_{\text{BC}}^{\text{FM}}$. Please refer to Appendix G for detailed pretraining hyperparameters. After pretraining, the backbone becomes an embodiment-agnostic foundation capable of rapid adaptation across heterogeneous robots. To deploy this model on novel domains with new hardware configurations $h_{\text{new}}$, we propose a lightweight two-step adaptation procedure:

**(1) Prompt warm-up.** We introduce new sets of learnable prompt $p_{\text{new}} \in \mathbb{R}^k$ for $h_{\text{new}}$. The prompt is first warmed up while keeping the pretrained weights frozen. By doing so, prompts are projected to exploit pretrained embodiment-agnostic features, offering good starts for next-round joint training.

**(2) Joint policy adaptation.** Then, we jointly optimize both the backbone and the warmed-up prompt, jointly adapt to new domains. This two-stage process first lets $p_{\text{new}}$ encode the hardware-specific setups of $h_{\text{new}}$, and then finetunes the full policy for effective specialization, sharing the same philosophy used to adapt LLMs to VLMs (Liu et al., 2023a; Li et al.).

**Custom learning rate (LR)**. A key stabilization technique in both pretraining and adaptation is the use of a reduced learning rate for the soft prompts as well as for the vision–language modules that respond for encoding visual and linguistic inputs. This adjustment reduces the risk of catastrophic drift from pretrained representations, an issue also noted by (Reuss et al., 2025; Driess et al., 2025), leading to smoother optimization during pretraining and more reliable specialization when adapting to novel embodiments. It effectively bridges the general knowledge encoded in vision–language models with the fine-grained spatial localization and action grounding required by VLA models.

### 4.2.2 ENHANCED DATA PROCESSING

**Aligned action representation.** Actions are the core supervision signals for VLA models, with their quality directly shaping training outcomes. Therefore, we standardize the action space into end-effector (EEF) pose representation comprising: (1) the Cartesian EEF xyz position, (2) the absolute EEF rotation encoded using the Rotate6D representation (Zhou et al., 2019) to avoid the discontinuities inherent in Euler angles and quaternion representations, and (3) the discretized binary state of the gripper. The position and rotation are optimized using mean-squared-error (MSE) loss, while the gripper state is optimized with binary-cross-entropy (BCE) loss. This ensures consistency across embodiments, providing robust supervision for generalizable policy learning.

Table 1: The ablation path for each components in Section 4. We evaluate the pretraining (PT) validation error and adaptation (AD) success rates on Simpler-WidowX benchmark (Li et al., 2025b). Green, Red and Gray denote positive, negative, moderate effects, respectively. **Bold** scores are SOTA results. We can see that naively training on heterogeneous data leads to degradation. Also, as validation error decreases during pretraining, the adaptation success rate increases progressively, demonstrating a strong correlation between the two. Therefore, we use the validation error as a proxy for pretraining performance throughout this paper. It is evident that each components in Section 4 contributes to positive improvements for pretraining.

| Type | Improvements | Val Error (PT) | Acc (AD) |
|---|---|---|---|
| Baseline Model (w/o PT) | Florence-base + Standard DiT-base | - | 4.1 |
| Pretraining Technique (Section 4.2.1) | +Custom LR (w/o PT) | - | 39.6 (+35.5) |
| | +Heterogeneous PT | 0.11 | 25.0 (-14.6) |
| Data Processing (Section 4.2.2) | +Action alignment | | |
| | +Intension abstraction | 0.077 (-0.033) | 50.0 (+25.0) |
| | +Balanced data sampling | | |
| Architecture Design (Section 4.1) | +Replace DiT with Transformer encoder | 0.071 (-0.006) | 47.9 (-2.1) |
| | +Encoding pipeline | 0.053 (-0.018) | 64.6 (+16.7) |
| | +Soft-prompt | 0.041 (-0.012) | **73.8** (+9.2) |
| | +Scaling up | 0.032 (-0.009) | **89.6** (+15.8) |
| Finetuning Technique (Section 4.2.1) | +Two-step adaptation | 0.032 | **95.8** (+6.2) |

**Intention abstraction through temporal downsampling.** While low-level action trajectories provide the precise manipulation signals required for deployment, they are often too fine-grained and may contain lots of noisy movements due to human randomness, thus are not suitable for achieving high-level grounding and intention modeling for pretraining. To mitigate this issue, we temporally downsample demonstrations to construct abstract representations of action intentions. Concretely, rather than predicting the full end-effector pose at every time step, the pipeline is designed to generate a sequence of 30 anchor points that summarize the intended trajectory over the next 4 seconds. The choice of the 4-second prediction window is based on extensive preliminary experiments, with detailed results provided in the Appendix D.4.

**Balanced data sampling strategy.** In contrast to the common round-robin data sampling strategy (Wang et al., 2024b), we observe that stable training requires a carefully designed data shuffling pipeline. We shuffle samples not only across different domains but also across trajectories within each domain, ensuring exposure to a diverse and balanced data mixture at every iteration. This effectively mitigates distributional bias and reduces overfitting to dominant domains, facilitating smoother convergence during large-scale pretraining.

## 4.3 IMPLEMENTATION DETAILS OF X-VLA-0.9B

In this paper, we build a 0.9B-parameter instantiation of X-VLA using heterogeneous embodied data sources, as illustrated in Fig. 3. Specifically, X-VLA adopts Florence-2-Large (Xiao et al., 2024) as its VLM encoder, given its strong performance in visual grounding and general vision language reasoning capabilities that are essential for embodied tasks (Reuss et al., 2025). X-VLA-0.9B employs a standard Transformer backbone with 24 layers and a hidden size of 1024 for action generation. A key component of the framework is the soft prompt, whose length is set to 32, which is determined through systematic scaling experiments across model sizes (Fig. 6). Additional pretraining and post-training details are provided in Appendix G.

## 5 EXPERIMENTS

In this section, we conduct extensive experiments to investigate 1. Does *X-VLA* exhibit scaling properties along model size, data diversity, and data scale? 2. Can *X-VLA* specialize to novel domains with varied characteristics? 3. Do the soft prompts capture meaningful representations that reflect the heterogeneity of mixed data sources?

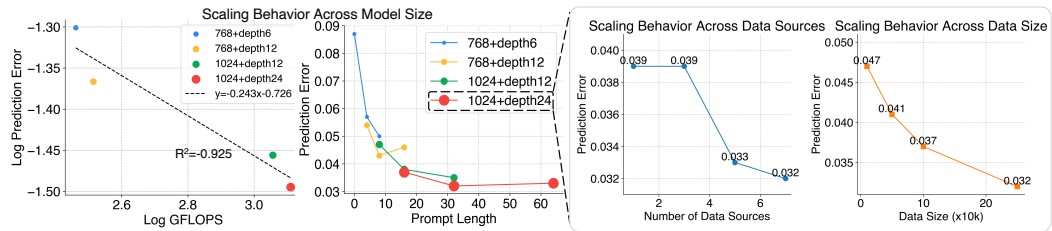

Figure 6: With increased compute, data diversity, and data volume, *X-VLA* can output reduced validation prediction error, which can lead to enhanced adaptation performance as discussed by Tab. 1.

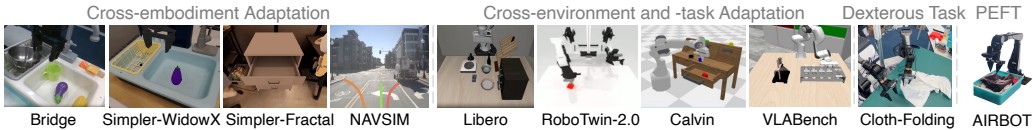

Figure 7: The evaluated setups in adaptation experiments, to evaluate different axes of adaptation capabilities.

## 5.1 SCALING EXPERIMENTS

First, we study the scaling behavior of *X-VLA* along three axes: (1) model capacity, (2) data diversity, and (3) data volume. As shown in Tab. 1, prediction errors observed during pretraining are strongly correlated with downstream adaptation performance. Therefore, we adopt the $\ell_1$ error between predicted actions (after flow-matching denoising) and ground-truth actions on held-out validation sets as our primary evaluation metric. The results are summarized in Fig. 6, with additional training details presented in Appendix G. Notably, even at the largest tested configuration, *X-VLA-0.9B* (hidden size 1024, 24 Transformer blocks), trained on 290K episodes from 7 data sources, the scaling trend shows no sign of saturation. This indicates that further increases along these three axes could yield additional performance gains. Due to resource constraints, we adopt the largest configuration, as the default model for subsequent experiments.

## 5.2 ADAPTATION EXPERIMENTS

We present one of the most comprehensive validation studies to date, evaluating *X-VLA-0.9B* across 6 simulation environments and 3 real-world robotic platforms. See Appendix D for more results.

**Simulation benchmarks.** We evaluate on Libero (Liu et al., 2024a), Simpler (Li et al., 2025b), VLABench (Zhang et al., 2024a), RoboTwin-2.0 (Chen et al., 2025), Calvin (Mees et al., 2022) and NAVSIM (Dauner et al., 2024). These 6 benchmarks encompass hundreds of evaluation setups, spanning single-arm, bi-manual robotic systems, autonomous driving and assessing diverse axes of generalization, including cross-embodiment, cross-environment, and cross-task adaptation. Across **FIVE** benchmarks, we establish a new SOTA, achieving substantial improvements over aggregated prior models. Remarkably, it attains over 90% success rates on several benchmarks, *e.g.*, Simpler-WidowX (96%), Libero (98%), and Calvin-1st stage. To the best of our knowledge, no prior model has reported such comprehensive evaluation paired with consistently significant gains, underscoring the superior performance of *X-VLA-0.9B*, which can become a strong baseline for future research to develop advanced models (please refer to Appendix H for details).

**Real-world experiments.** We also evaluate *X-VLA-0.9B* on physical robotic platforms follow the BridgeData-v2 benchmark (Walke et al., 2023), the evaluation details can be found in Appendix J and the results are reported in Fig. 8. Our *X-VLA* surpass other baselines across all five tasks, each for testing distinct axis of capability, demonstrating the superior adaptability of our *X-VLA*.

**Dexterous cloth-folding task.** We introduce a challenging dexterous cloth-folding task that requires smoothing highly disordered cloth and folding it neatly. To support this effort, we build a high-quality cloth-folding dataset on the bi-manual Agilex platform, termed Soft-Fold, which consists of 1,200 trajectories collected through a carefully designed pipeline. A detailed description of both the task and the dataset is provided in Appendix F. Importantly, we will **release** the dataset to facilitate future research in dexterous manipulation. Leveraging this dataset for adaptation, our *X-VLA-0.9B* model achieves a throughput of nearly 100% success rate and 33 completed folds per

Table 2: Comparison of specialize and generalize models on simulation benchmarks

| Methods | Size | Simpler | | | LIBERO | | | | | Calvin | RoboTwin-2.0 | | VLABench | NAVSIM |
|---|---|---|---|---|---|---|---|---|---|---|---|---|---|---|
| | | VM | VA | WidowX | Spatial | Object | Goal | Long | Avg | ABC → D | Easy | Hard | Avg. PS | PDMS |
| LBP (Liu et al., 2025a) | 0.2B | - | - | - | - | - | - | 88.6 | - | - | - | - | - | - |
| MoDE (Reuss et al., 2024) | 0.4B | - | - | - | - | - | - | 94.0 | - | 4.01 | - | - | - | - |
| SuSIE (Black et al., 2024) | 1B | - | - | - | - | - | - | 76.3 | - | 2.69 | - | - | - | - |
| GHIL-Glue (Hatch et al., 2025) | 1B | - | - | - | - | - | - | - | - | 3.69 | - | - | - | - |
| SpatialVLA (Qu et al., 2025) | 4B | 75.1 | 70.7 | 42.7 | 88.2 | 89.9 | 78.6 | 55.5 | 78.1 | - | - | - | - | - |
| TraceVLA (Zheng et al., 2024b) | 7B | 46.2 | 49.1 | - | 84.6 | 85.2 | 75.1 | 54.1 | 74.8 | - | - | - | - | - |
| ThinkAct (Huang et al., 2025) | 7B | 71.5 | 65.1 | 43.8 | 88.3 | 91.4 | 87.1 | 70.9 | 84.4 | - | - | - | - | - |
| FPC-VLA (Yang et al., 2025) | 7B | 78.0 | 65.8 | 64.6 | 86.2 | 87.0 | 92.0 | 82.2 | 86.9 | - | - | - | - | - |
| MemoryVLA (Shi et al., 2025a) | 7B | 77.7 | 72.7 | 71.9 | 98.4 | 98.4 | 96.4 | 93.4 | 96.7 | - | - | - | - | - |
| Octo (Octo Model Team et al., 2024) | 0.1B | 16.8 | 1.10 | 23.4 | 78.9 | 85.7 | 84.6 | 51.1 | 75.1 | - | - | - | - | - |
| GR-1 (Wu et al., 2023) | 0.2B | - | - | - | - | - | - | - | - | 3.06 | - | - | - | - |
| Seer (Tian et al., 2025) | 0.3B | - | - | - | - | - | - | 87.7 | - | 4.28 | - | - | - | - |
| UniAct (Zheng et al., 2025) | 0.5B | - | - | - | 77.0 | 87.0 | 77.0 | 70.0 | 77.8 | - | - | - | - | - |
| RDT (Liu et al., 2025b) | 1B | - | - | - | - | - | - | - | - | - | 34.5 | 13.7 | - | - |
| FLOWER (Reuss et al., 2025) | 1B | - | - | 40.0 | 97.1 | 96.7 | 95.6 | 93.5 | 95.7 | 4.53 | - | - | - | - |
| SmolVLA (Shukor et al., 2025) | 2B | - | - | - | 93.0 | 94.0 | 91.0 | 77.0 | 88.8 | - | - | - | - | - |
| GR00T-N1 (NVIDIA et al., 2025) | 3B | 45.0 | 48.4 | - | 94.4 | 97.6 | 93.0 | 90.6 | 93.9 | - | - | - | 39.7 | - |
| π0 (Black et al., 2025b) | 3B | 58.8 | 56.8 | 27.8 | 96.8 | 98.8 | 95.8 | 85.2 | 94.1 | - | 46.4 | 16.4 | 37.8 | - |
| π0+FAST (Pertsch et al., 2025) | 3B | 61.9 | 60.5 | 39.5 | 96.4 | 96.8 | 88.6 | 60.2 | 85.5 | - | - | - | 34.1 | - |
| OpenVLA (Kim et al.) | 7B | - | - | 8.30 | 84.7 | 88.4 | 79.2 | 53.7 | 76.5 | - | - | - | - | - |
| OpenVLA-OFT (Kim et al., 2025) | 7B | 63.0 | 54.3 | 31.3 | 97.6 | 98.4 | 97.9 | 94.5 | 97.1 | - | - | - | - | - |
| DD-VLA (Liang et al., 2025) | 7B | 71.2 | 64.1 | 49.3 | 97.2 | 98.6 | 97.4 | 92.0 | 96.3 | - | - | - | - | - |
| UniVLA (Wang et al., 2025a) | 9B | - | - | 69.8 | 95.4 | 98.8 | 93.6 | 94.0 | 95.4 | 4.41 | - | - | - | 81.7 |
| Maximum of Existing SOTA | - | 78.0 | 72.7 | 71.9 | 98.4 | 98.8 | 97.9 | 94.5 | 97.1 | 4.53 | 46.4 | 16.4 | 39.7 | 81.7 |
| **X-VLA (Ours)** | **0.9B** | **80.4** | **75.7** | **95.8** | 98.2 | 98.6 | 97.8 | **97.6** | **98.1** | 4.43 | **70.0** | **39.0** | **51.1** | **87.3** |

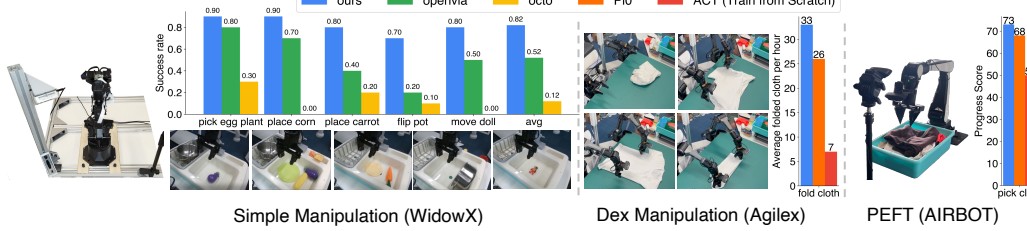

Figure 8: Real-World Evaluation Results. We evaluate our *X-VLA* model on three distinct real-world embodiments, each under specific task setups, including simple manipulation, dexterous manipulation, and fast adaptation experiments using PEFT techniques. See Appendix J for for details.

hour—comparable to the closed-source $\pi_0$-folding model (Black et al., 2025a), which is presumably trained on substantially larger and higher-quality datasets. For a fair comparison, we finetuned $\pi_0$-base and ACT model on Soft-Fold, but it failed to match the throughput of *X-VLA-0.9B*, underscoring the strong dexterous manipulation capabilities of our model. Additional qualitative results are provided in Appendix F and showcased in our web demos: website.

**Parameter efficient finetuning (PEFT) experiments.** To evaluate whether the pretrained *X-VLA-0.9B* backbone encodes embodiment-agnostic features and can be efficiently adapted to new settings, we adopt PEFT techniques such as Low-Rank Adaptation (LoRA) (Hu et al., 2022). We test adaptation on three benchmarks: Libero, Simpler-WidowX, and a cloth-pick task on AIRBOT, a real-world embodiment unseen during pretraining. Tab. 3 and Tab. 8 show that with only 9M tunable parameters (about 1% of the full model), the backbone can be steered into a strong domain-specialized model, achieving 93% and 54% success rates on Libero and Simpler-WidowX benchmarks, respectively. These scores are comparable to fully finetuned models, e.g., $\pi_0$ (Black et al., 2025a) achieve 94.2% and 55.7% on Libero and Simpler-WidowX, respectively.

## 5.3 IN-DEPTH ANALYSIS

We further demystify the effects of soft prompts through both qualitative and quantitative results, examining whether the proposed soft-prompt mechanism can effectively absorb meaningful domain-specific knowledge from heterogeneous datasets.

**Qualitative experiments.** We visualize the soft prompts learned after pretraining on the mixed data recipe (Fig. 3) using T-SNE (Maaten & Hinton, 2008). Each dot in Fig. 9 corresponds to a token from the soft prompt sequence of a specific data source. Since each data source includes multiple tokens

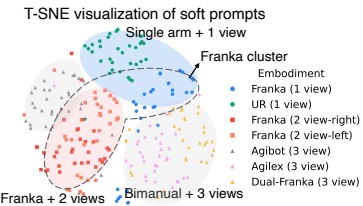
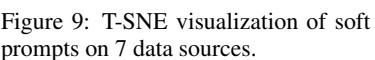

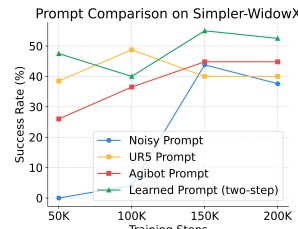

| Methods | $\pi_0$ | Ours-Lora |
|---|---|---|
| **#Param** | 3B | 9M |
| Libero-Spatial | 96.8 | $95.8 \pm 0.4$ |
| Libero-Object | 98.8 | $96.3 \pm 0.3$ |
| Libero-Goal | 95.8 | $95.2 \pm 0.8$ |
| Libero-Long | 85.2 | $83.7 \pm 0.5$ |
| Simpler-WidowX | 55.7 | 54.2 |

Figure 9: T-SNE visualization of soft prompts on 7 data sources.

Figure 10: Comparison of different prompts on PEFT.

Table 3: PEFT performance comparison across benchmarks.

to encode embodiment heterogeneity, jointly visualizing all tokens across embodiments intuitively reflects both intra- and inter-embodiment relationships. Fig. 9 reveals that the prompts form well-structured clusters that align closely with different hardware configurations, indicating that they successfully capture embodiment-specific information. More excitingly, the two Franka setups (with left and right views) derived from Droid data are intermingled rather than separated, as they only differ in their designated main view. This observation suggests that soft prompts do not merely partition data sources in a brute-force manner but instead leverage cross-embodiment similarities.

**Quantitative experiments.** Further, we evaluate how pretrained soft prompts facilitate efficient adaptation to WidowX, an single-arm robot unseen in pretraining. We conduct PEFT experiments on Simpler, comparing four settings: (1) randomly initialized soft prompts kept frozen, (2) pretrained prompts from dual-arm platform (e.g., AgiBot) kept frozen. (3) pretrained prompts from another single-arm platform (e.g., UR5) kept frozen, and (4) soft prompts adapted with our two-step adaptation mechanism. In Fig. 10, it's no surprise that learned prompts converge faster and finally reach higher success rates, whereas random prompts lead to slower adaptation and degraded performance. However, it's good to see that the frozen pretrained prompts offer strong transfer benefits in early stage due to the partial similarity between UR5 and WidowX, although the inevitable domain gap limits the final performance. Also, the AgiBot prompts offer subpar benefits at the beginning of training compared to UR5 prompts, largely due to the substantial embodiment mismatch between single arm and bi-manual arms. This highlights a promising avenue for cross-embodiment transfer: with pretraining on more diverse robotic platforms, soft prompts may enable zero-shot/few-shot generalization by retrieving prompts aligned with the closest hardware setups.

## 6 CONCLUSION

In this paper, we introduce *X-VLA*, a generalist Vision-Language-Action framework capable of operateing across heterogeneous robotic platforms. Through a carefully designed training pipeline, adaptation methods, and enhanced data processing, our largest model *X-VLA-0.9B* achieves SOTA performance across a broad spectrum of benchmarks, setting new records with substantial gains over hundreds of evaluation setups. Remarkably, even with minimal tunable parameters, *X-VLA-0.9B* delivers results competitive with fully finetuned SOTA models. Importantly, empowered by *Soft Prompt* mechanism, *X-VLA* exhibits scalable training trends along all three axes including model size, data diversity and data volume without sighs of saturation even at our largest test configuration (0.9B parameters, 290K episodes, 7 data sources). This highlights its potential for further scaling to larger models and datasets, paving the way toward more powerful VLA models. Limitations and future works are discussed in Appendix C.

## ACKNOWLEDGMENTS

This work was supported by funding from the National Key R&D Program of China (2022ZD0160201), Shanghai AI Lab, Wuxi Research Institute of Applied Technologies, Tsinghua University (Grant No. 20242001120), Beijing Academy of Artificial Intelligence (BAAI), and Horizon Robotics. We thank Wencong Zhang for the help on robot maintenance, Yiming Meng for the help on surveying simulation benchmarks, and Yiming Chen for the help on data collection.

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

## A    LLM Usage and Ethics Statement

In this paper, we employed Large Language Models (LLMs) **solely** for polishing the writing. No parts of the technical content, experimental results, or conclusions were generated by LLMs.

A potential ethical concern is the use of large-scale pretraining data, which may contain privacy-sensitive information or embedded biases. To mitigate this, we use exclusively open-sourced robotics datasets (Bu et al., 2025; Wu et al., 2024; Khazatsky et al., 2024; O'Neill et al., 2024), all of which have undergone peer review and are widely adopted in the research community. We believe this substantially reduces the risk of privacy violations or biased data influencing our results.

Nevertheless, we encourage future researchers to exercise caution when curating data for training large-scale robotics models, particularly by filtering privacy-sensitive content and addressing potential biases to ensure responsible and fair deployments of embodied AI systems.

## B    Related Work

**Vision-Language-Action Models.** Developing agents that can interact with the physical world requires integrating three essential modalities: visual perception to understand the environment, language comprehension to interpret task instructions, and action generation to produce executable control signals. Research on Vision-Language-Action (VLA) models (Black et al., 2025a; NVIDIA et al., 2025; Zheng et al., 2025; Kim et al.) has focused on unifying these modalities to enable embodied agents to perform complex tasks conditioned on natural language commands and visual observations. These models are typically built upon Vision-Language Models (VLMs) (Team et al., 2024; Achiam et al., 2023; Li et al., 2024a; Xiao et al., 2024), which are pretrained on large-scale vision–language corpora. By inheriting strong visual grounding and generalist reasoning capabilities from VLMs, VLA models achieve impressive results on diverse manipulation tasks. More recently, researchers have recognized the inherent gap between general-purpose vision–language reasoning and embodied task requirements (Qu et al., 2025; Black et al., 2025a; Mu et al., 2023). To address this, various approaches have been explored, such as incorporating embodiment-specific priors (e.g., 3D spatial grounding (Qu et al., 2025; Shi et al., 2025a), instruction-following (Zheng et al., 2024a), historical reasoning (Shi et al., 2025a)) via modality injection (Qu et al., 2025), scaling up domain-relevant datasets (Black et al., 2025a; Li et al., 2024b), adding extra supervision, or designing specialized models (Shi et al., 2025a). Nevertheless, due to the inherent limitations of current VLMs, these strategies often fall short of achieving the level of generalized reasoning required for embodied tasks with complex visual inputs. In this paper, we demonstrate that a simple yet effective modification of the input streams can better harness the generalization potential of VLMs, leading to significant performance gains.

**Heterogeneous Pretraining.** Training on large-scale datasets has been a key factor of recent progress in embodied foundation models (Black et al., 2025b; Kim et al.). However, robotics data available for large-scale pretraining often exhibit strong heterogeneity, not only in action spaces but also in hardware setups (O'Neill et al., 2024). To address this, Wang et al. (2024b) proposed pretraining a standard Transformer on heterogeneous data mixtures with carefully designed architectural modifications, demonstrating promising scaling behavior and transferability. More recently, researchers have observed that pretrained VLMs already possess strong generalization ability in handling diverse vision–language inputs across domains. Consequently, the focus has shifted toward resolving heterogeneity in action spaces (Zheng et al., 2025; NVIDIA et al., 2025; Zawalski et al., 2024). Beyond manually reshaping action spaces (Liu et al., 2025b) or modifying model architectures to accommodate heterogeneous action labels, several approaches have been proposed to align actions at the semantic level through latent action modeling or representation learning (Ye et al., 2024; Zheng et al., 2025; Li et al., 2024b; 2025a). Nevertheless, learning policies from heterogeneous data sources requires more than aligning action labels, but embodiment-specific and proprioceptive-aware reasoning, since variations in hardware setups directly affect how observations map to actions. Simply feeding heterogeneous data into a shared backbone without explicitly modeling these embodiment-specific factors often leads to unstable training and poor cross-domain generalization. In this paper, we introduce a soft-prompt mechanism that explicitly absorbs embodiment-specific variability while preserving a shared backbone for general reasoning. By as-

sociating each hardware configuration with learnable prompt embeddings, the model can flexibly capture domain-specific knowledge, thus enabling stable large-scale pretraining.

**Soft Prompt Learning**. The concept of soft prompts was originally introduced in the NLP community as a parameter-efficient alternative to full model finetuning. Instead of updating all parameters of a pretrained model, a small set of learnable embeddings are prepended to the input sequence and optimized for downstream tasks (Lester et al., 2021). This approach has proven highly effective in adapting large language models to diverse tasks with minimal additional parameters, inspiring extensive research on prompt-based transfer learning across modalities (Li & Liang, 2021; Wang et al., 2022; Liu et al., 2024b). Building on this foundation, soft prompts have been extended to multimodal and multi-task learning settings (Liu et al., 2023c). When combined with the philosophy of meta-learning (Park et al., 2024; Gordon et al., 2019), soft prompts can serve as lightweight carriers of domain- or task-specific information. By injecting learnable embeddings that guide the backbone without overwriting its pretrained representations, they provide a flexible and scalable mechanism to handle domain heterogeneity (Wang et al., 2023). In this work, we adopt the soft prompt paradigm for robotics, where heterogeneity arises from embodiment differences such as hardware configurations and action spaces. We demonstrate that soft prompts can effectively absorb embodiment-specific variability, enabling the backbone to focus on learning an embodiment-agnostic generalist policy.

## C    LIMITATIONS AND FUTURE WORKS

In this section, we discuss the limitations of our work and outline potential directions for future research.

**Scaling *X-VLA* with broader data and model sizes.** While *X-VLA-0.9B* achieves strong performance, its scale remains modest compared to large foundation models in the vision–language and language domains. This limitation stems primarily from computational constraints and the limited availability of high-quality robotics data. Despite our efforts to collect and curate open-source datasets (Wu et al., 2024; O'Neill et al., 2024; Bu et al., 2025), the diversity and scale of current robotics corpora still fall short of those in language or vision–language domains. Scaling *X-VLA* to larger capacities, either by expanding the backbone or leveraging stronger pretrained VLMs, and training on broader, more diverse robotics datasets could further enhance generalization and robustness. Such extensions also raise open questions about the scaling laws of VLA models and how embodiment-specific variability interacts with model capacity.

**Enhancing supervision signals for large-scale robotics pretraining.** Despite our efforts to mitigate heterogeneity across data sources and to align action spaces for generalized knowledge learning, the supervision provided by low-dimensional action labels remains inherently limited in information content. These labels, while essential for direct control, capture only a narrow view of the underlying task structure and often fail to convey higher-level reasoning, intent, or multi-step dependencies. In this work, we show that a simple temporal downsampling strategy can help abstract action intentions and thereby facilitate more efficient pretraining. However, such heuristics only partially address the problem, as they do not fundamentally enrich the supervision. Future directions include incorporating richer supervisory signals such as 3D spatial reasoning cues, physical dynamics or intermediate subgoal annotations. Another promising avenue is leveraging self-supervised objectives from raw input streams to complement sparse action labels, thereby enhancing representation learning and improving scalability in heterogeneous, real-world robotics settings.

**Towards a generalist model seamlessly deployed to downstream tasks.** Our *X-VLA* demonstrates superior performance across various downstream tasks, showing strong adaptability under fine-tuning and efficient specialization. However, realizing the vision of a truly generalist embodied model that can be seamlessly deployed to arbitrary downstream tasks without additional engineering or retraining remains an open challenge. Currently, deployment still relies on embodiment-specific adaptation, typically involving the collection of a small number of demonstrations for post-training. While these strategies are lightweight compared to full retraining, they nonetheless introduce overhead and prevent the model from serving as a true plug-and-play solution in real-world applications. Moreover, the dependence on embodiment-specific data becomes problematic when scaling to platforms where high-quality demonstrations are scarce, expensive, or risky to collect. Future research should therefore focus on approaches that move closer to seamless deployment. Based

on the empirical finding in this paper, a promising direction include exploring unified embodiment representations: incorporating explicit embodiment-agnostic abstractions (e.g., universal kinematic descriptors, physics-informed priors) to reduce reliance on task-specific adaptation.

# D   MORE RESULTS

In this section, we present additional results that highlight the strengths of our approach. Specifically, we report: (1) comparisons between our model and alternative architectural designs, (2) evaluations under cross-embodiment joint training, (3) evaluations in data-constrained settings, (4) ablation study on prediction window during pretraining and (5) ablation study on parameter-efficient finetuning methods

| | DiT | MM-DiT | $\pi_0$-Style | **Ours** |
|---|---|---|---|---|
| Validation Error | 0.077 | 0.140 | 0.056 | **0.041** |

Table 4: Comparison of backbone architectures on validation error. *X-VLA* achieves the lowest error while maintaining stable training.

| | Libero-Long | Simpler-WidowX | Calvin |
|---|---|---|---|
| Single-domain FT | 97.6 | 96.0 | 4.42 |
| Multi-domain FT | **98.1** | 93.8 | 4.32 |

Table 5: Joint adaptation to multiple embodiments. Multi-domain finetuning achieves performance comparable to, and in some cases exceeding, single-domain finetuning, demonstrating the scalability of *X-VLA* to heterogeneous deployment settings.

## D.1   ALTERNATIVE ARCHITECTURAL DESIGNS

In this section, we present additional results from alternative architectural designs explored during development. While our final model adopts the *X-VLA* pipeline, we also implemented several commonly used backbone architectures for comparison. These baselines were evaluated under identical experimental settings, consistent with the preliminary experiments described in Appendix I.

**Standard DiT Decoder.** A direct application of the Diffusion Transformer (DiT) decoder (Peebles & Xie, 2023) that generates actions conditioned on multimodal features extracted by pretrained vision–language encoders. This is the most straightforward extension of DiT to embodied settings.

**Standard MM-DiT Decoder.** A multimodal variant of DiT that allocates separate parameters for different input modalities and integrates them through attention (Esser et al., 2024). We isolate the action modality from visual–language inputs. Although this design attempts to reduce the semantic gap across modalities, it often destabilizes training and leads to inferior results on heterogeneous datasets and downstream adaptation.

$\pi_0$**-Style Decoder.** Following (Black et al., 2025b), this design employs a parallel MLP-Mixer (Tolstikhin et al., 2021)–based action module alongside a pretrained VLM for vision–language processing. This leverages the compact nature of action inputs, which can be effectively represented with dense feed-forward networks, but comes at the cost of added architectural complexity.

The comparative results across these backbones are summarized in Tab. 4, where our *X-VLA* consistently achieves the best validation performance while maintaining stable optimization dynamics.

## D.2   POTENTIAL TO BUILD CROSS-EMBODIMENT GENERALIZED POLICY

Empowered by soft prompts, *X-VLA* enables efficient and stable training on heterogeneous datasets, effectively absorbing domain variations and fostering embodiment-agnostic policy learning. Building on this capability, we show that *X-VLA* can be adapted not only to a single novel embodiment but also to multiple embodiments simultaneously through joint finetuning on demonstrations from diverse data sources. Concretely, we conduct joint finetuning experiments on a mixture of downstream datasets including Libero, BridgeData, and Calvin-ABC, which include two distinct embodiments and three hardware setups for both data collection and deployment. After joint finetuning using the same training recipe as other finetuning experiments in Appendix H, we report the results in Table 5.

Table 5 shows the multi-domain adaptation results. *X-VLA* maintains consistently strong performance across all evaluated embodiments when adapted jointly, demonstrating its ability to scale beyond single-domain specialization. Interestingly, joint adaptation not only preserves performance

within each domain but in some cases even improves success rates compared to single-domain fine-tuning, suggesting positive cross-domain transfer. This indicates that the soft-prompt mechanism not only absorbs embodiment-specific variations but also enables complementary knowledge sharing across heterogeneous embodiments.

## D.3 DATA-EFFICIENT ADAPTATION

| # demos | Libero-Spatial | Libero-Object | Libero-Goal | Libero-Long | Libero-Avg |
|---|---|---|---|---|---|
| 50 (Full & Default) | 96.6 | 95.4 | 95 | 84.2 | 92.8 |
| 10 | 95.2 | 94.2 | 93.6 | 81.5 | 91.1 |

Table 6: Data-efficient adaptation performance of PEFT finetuned *X-VLA-0.9B* on Libero under limited demonstrations. Even with only 10 demonstrations, the model maintains strong performance.

In this section, we investigate whether the learned embodiment-agnostic backbone can be efficiently adapted to novel embodiments under limited supervision. To this end, we finetune *X-VLA-0.9B* in a PEFT setup on Libero-Goal using only a small number of demonstrations. As shown in Table 6, the model achieves a 92.8% success rate with 50 demonstrations, and remarkably still retains a strong 91.1% success rate with only 10 demonstrations. These results highlight the data efficiency of our two-step adaptation procedure, showing that the pretrained backbone, together with soft prompts, serves as a strong prior that enables effective specialization even under extreme data scarcity.

## D.4 ABLATION STUDY ON THE PREDICTION WINDOW DURING PRETRAINING

The prediction window for pretraining on the curated datasets is selected through a careful grid search under a small-scale preliminary setup. Specifically, we pretrained a 0.3B version of X-VLA (Florence-base and 100M self attention block) on our curated data recipe for 100K iterations and evaluated the resulting model via PEFT on Simpler-WidowX, since validation loss under different prediction targets are not directly comparable. As shown in Tab. 7, a 4-second window emerges as a reasonable choice: it captures the intended action trajectory clearly while retaining sufficient future dynamics without excessive loss of detail.

| prediction window | Success Rate |
|---|---|
| 1s | 0 |
| 2s | 8.3 |
| 4s | 29.16 |
| 8s | 27.08 |

Table 7: Ablation study on the prediction window during pretraining.

| # Param | Success Rate |
|---|---|
| Prompt only (32K) | 0 |
| + Linear head ( 70K) | 8.3 |
| + LoRA (9M) | 54.2 |
| + Unfreeze last layer (25M) | **68.9** |

Table 8: Ablation study on parameter-efficient finetuning methods.

## D.5 ABLATION STUDY ON PARAMETER-EFFICIENT FINETUNING METHODS

In addition to our standard PEFT approach, which combines two-phase soft-prompt tuning with LoRA finetuning, we conducted an additional experiment using standard prompt tuning, where we also unfroze several last layers of the backbone to increase the number of learnable parameters and examine the effect of tunable parameter count. The results of this study are reported in Tab. 8. These findings indicate that the number of tunable parameters plays an important role in effectively adapting X-VLA to specific domains.

## E  FAILURE ATTEMPTS FOR ABSORBING HETEROGENEITY

The core motivation of this paper is to explore strategies for mitigating heterogeneity across mixed data sources and to develop a generalist, embodiment-agnostic policy. Inspired by the philosophy of meta-learning (Gordon et al., 2019), we initially approached this problem from the perspective of heterogeneous parameter learning. Concretely, we assigned distinct parameter sets for each domain, with the expectation that these domain-specific parameters could absorb domain variations while the

Figure 11: The illustration of our proposed Soft-Fold datasets.

shared backbone distilled generalizable knowledge across domains. Ultimately, we found that our proposed soft-prompt mechanism provides an effective solution to this challenge. In this section, we present two of our unsuccessful attempts, with the aim of highlighting practical pitfalls and inspiring future work in this direction.

**Heterogeneous Low-rank Adapter**. Beyond soft prompts, we explored the integration of other parameter-efficient learning methods into heterogeneous pretraining. Specifically, we experimented with Low-Rank Adaptation (LoRA)-style modules (Hu et al., 2022), where domain-specific adapters were introduced in parallel with the shared backbone. Our intuition was that these adapters could capture embodiment-specific variations with efficient parameterization, and meanwhile the main backbone encodes embodiment-agnostic features. However, we observed that the additional adapters often conflicted with the optimization dynamics of the backbone, leading to **instability** and **degraded generalization** across domains.

**Heterogeneity-guided MoE framework**. We also experimented with a mixture-of-experts (MoE) approach, which has been widely used for scaling model capacity while controlling inference cost. MoE's sparse activation mechanism (**?**) has proven effective in multi-task learning (Pham et al., 2023), cross-domain learning (Zhang et al., 2024b), and multi-modal robotics behavior cloning (Reuss et al., 2024). Motivated by these successes, we designed a heterogeneity-guided routing strategy that aimed to activate experts based on embodiment-specific cues. Despite its theoretical appeal, we found that the router tended to **collapse**, consistently routing most inputs to only a few experts while leaving others underutilized, leading to wasted capacity and only marginal performance gains. To mitigate this, we give another try to introduce load-balancing regularization (Wang et al., 2024a), but the resulting rapid switching across experts often destabilized optimization and degraded overall training dynamics.

## F    SOFT-FOLD: SUPERIOR DEXTEROUS MANIPULATION MODEL WITH A HIGH-QUALITY CLOTH FOLDING DATASET

We provide qualitative results about our finetuned dexterous manipulation model from the pretrained *X-VLA-0.9B* and introduce a high-quality cloth folding dataset: Soft-FOLD, as illustrated in Fig 11.

**Demonstration collecting strategy.** Humans can fold clothes casually and quickly, often using a wide variety of methods in a seemingly random manner. However, this variability poses significant challenges for robotic policy learning, since different folding strategies often correspond to distinct

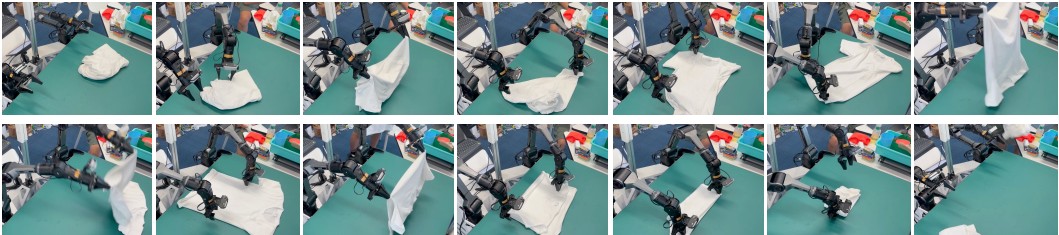

Figure 12: The folding progress of *X-VLA-0.9B*.

behavioral modes, and not all strategies are equally suitable for training. To reduce the inconsistency in human demonstrations, we decompose the folding task into two stages: (1) smoothing out the cloth from a highly disordered state, and (2) folding the smoothed cloth neatly. We find that the first stage is particularly challenging, as the disordered cloth exhibits highly random dynamics, requiring the policy to capture a universal strategy for unfolding. To address this, we collect demonstrations for stage I in a **repetitive** manner until meaningful keypoints, such as the two corners or two ends of the cloth emerge clearly. At that point, we employ swinging motions to complete the smoothing stage and then transition to stage II. This is critical for cloth folding, as unstructured or randomly collected demonstrations in stage I can entangle policies in inconsistent behaviors, leading to unstable learning dynamics and hindering progression to stage II. For stage II, the data collection becomes far more easier, as the cloth behaves less randomly after smooth-out. On average, one full folding episode takes about 1.5 minutes, with one hour of collection yielding 20–25 episodes, including time for resetting and discarding failed attempts. The final dataset includes 1,200 episodes, as shown in Fig. 11.

**DAgger-style data collection**. To train long-horizon dexterous tasks such as cloth folding with limited episodes, we find it essential to adopt a DAgger-style data collection strategy (Ross et al., 2011), a practice also noted by Hu et al. (2025). Concretely, we train ACT (Zhao et al., 2023) after every 100 collected episodes, identify its failure modes, and then collect targeted demonstrations to address these failures. This iterative refinement enables us to achieve cloth-folding performance comparable to that of closed-source models that are likely trained on substantially larger datasets, using only 1,200 episodes.

**Qualitative results of *X-VLA-0.9B*.** Here, we visualize a complete folding progress of our *X-VLA-0.9B* in Fig. 12. One complete folding covers diverse skills, such as the simple `Localization`, `Pick`, `Place` and high-dynamical `Swing` motion, demonstrating the challenging of the cloth-folding tasks.

## G    PRETRAINING DETAILS

The pretraining of *X-VLA-0.9B* was carried out on 64 NVIDIA A100 GPUs with a global batch size of 1024, costing approximately 4 days. The training followed a carefully tuned recipe to ensure stability and efficient convergence across heterogeneous datasets.

Tab. 9 summarizes the core hyperparameters used during pretraining. We adopt the AdamW optimizer with momentum parameters $\beta_1 = 0.9$ and $\beta_2 = 0.95$, a learning rate of $1 \times 10^{-4}$, and a weight decay of 0.01. Training was performed for 200K iterations with mixed-precision (bfloat16). For visual inputs, images are resized to $224 \times 224$ and augmented with mild perturbations using ColorJitter to improve generalization.

In addition to the optimizer configuration, a critical aspect of pretraining is balancing the heterogeneous datasets. Since different sources vary greatly in both scale and quality, we adopt a weighted sampling strategy combined with a carefully designed shuffling pipeline. As highlighted in Sec 4.2.1, this includes cross-domain and cross-trajectory shuffling, ensuring that the model is consistently exposed to a diverse and balanced mixture of samples at every iteration. We find this design crucial for stabilizing optimization and preventing domain overfitting during large-scale pretraining. Tab. 10 summarizes the data sources, available trajectories, and the sampling weights applied.

**Validation set construction.** We conduct open-loop validation experiments to monitor pretraining convergence and ensure fair comparisons across different architectures and methods. To guarantee that the validation loss serves as a clear and reliable proxy for downstream task performance, we carefully construct a dedicated validation set. Specifically, we sample trajectories from AGIBOT-beta (Bu et al., 2025) that are excluded from the training split, allowing us to better evaluate cross-embodiment knowledge sharing and generalization. The validation set spans 189 tasks, with three trajectories sampled per task. For evaluation, we report the average $\ell_1$ error between the predicted and ground-truth trajectories.

| Configuration | Value |
|---|---|
| Optimizer | AdamW |
| Batch size | 1024 |
| Learning rate | $1 \times 10^{-4}$ |
| Weight decay | 0.01 |
| Optimizer momentum | $\beta_1, \beta_2 = 0.9, 0.95$ |
| Training iterations | 200K |
| Model precision | bfloat16 |
| Image Resize | 224x224 |
| Image Augmentation | ColorJitter(0.2, 0.2, 0.2, 0) |

Table 9: Hyperparameters for pretraining.

| Data source | Num. traj | Sampling weight |
|---|---|---|
| AGIBOT | 141K | 0.4 |
| Droid-Left | 45K | 0.15 |
| Droid-Left | 45K | 0.15 |
| RoboMind-Franka | 19K | 0.1 |
| RoboMind-Dual-Franka | 2K | 0.03 |
| RoboMind-UR | 25K | 0.1 |
| RoboMind-Agilex | 11K | 0.07 |

Table 10: Sampling weights for heterogeneous data sources during pretraining.

## H  FINETUNING DETAILS

In this section, we provide additional training details for the adaptation experiments. Unless otherwise specified, the optimizer settings (AdamW with $\beta_1 = 0.9, \beta_2 = 0.95$), weight decay (0.01), model precision (bfloat16), and learning rate ($1 \times 10^{-4}$) are kept consistent with the pretraining stage. All models are adapted using our proposed two-step procedure: during the first 1,000 iterations, only the soft prompts and action heads are updated while all other parameters remain frozen; this is followed by a 1,000-iteration warm-up phase that gradually restores the learning rate to its default value for joint training.

Table 11 summarizes the benchmark-specific hyperparameters. For clarity, `Abs EEF` denotes the absolute end-effector position control interface, while `Rel XYZ + Abs Rotation` refers to relative Cartesian translation combined with absolute rotation. All rotations are parameterized using the 6D representation, and the gripper state is binarized and predicted via a sigmoid activation. To maximize knowledge transfer from the pretrained backbone, we adopt aligned action representations (`Abs EEF`) across most downstream benchmarks. However, in the Simpler-Google benchmark, where the camera setup is deliberately altered to test robustness against visual variation, we adopt the `Rel XYZ + Abs Rotation` control interface due to the sensitivity of absolute parameterizations to domain shifts in perception.

| Benchmark | Control Interface | Batch Size | Training Steps | Data Augmentation |
|---|---|---|---|---|
| CALVIN-ABC | Abs EEF | 128 | 60K | ColorJitter |
| LIBERO | Abs EEF | 128 | 60K | - |
| RobotWin-2.0 | Abs EEF | 128 | 60K | ColorJitter |
| VLA-Bench | Abs EEF | 128 | 60K | ColorJitter |
| BridgeData | Abs EEF | 128 | 60K | ColorJitter |
| FactalData | Rel XYZ + Abs Rotation | 256 | 50K | RandomResizeCrop + ColorJitter |
| SoftFold | Abs EEF | 256 | 400K | ColorJitter |
| PEFT experiments | Abs EEF | 128 | 40K | ColorJitter |

Table 11: Finetuning hyperparameters for each downstream benchmark. Settings follow pretraining defaults unless otherwise specified.

## I  TRAINING DETAILS FOR PRELIMINARY EXPERIMENTS

In this section, we provide additional details on the preliminary experiments. We adopt Florence-Base (Xiao et al., 2024) as the vision–language encoder and configure the backbone as a standard

DiT-Base (12 Transformer layers, hidden size 768, with AdaLN conditioning) to ensure comparability. Training is conducted on the curated heterogeneous data mixture using 8 NVIDIA A100 GPUs with a global batch size of 256 for 200K iterations. Unless otherwise specified, all remaining settings (optimizer, weight decay, augmentation, and shuffling strategy) are kept consistent with the pretraining setup described in Section G. Following we provide more implementation details about baseline methods.

**HPT-style Methods.** Following (Wang et al., 2024b), we implement a cross-attention–based resampler that maps domain-specific observations into a shared representation space before feeding them into the DiT decoder. Each domain is assigned its own resampler and a dedicated action head, while the core Transformer backbone remains shared across all domains. This design aims to mitigate observation heterogeneity while keeping the reasoning backbone general.

**Language Prompts.** In this setting, we provide embodiment-aware textual descriptions that encode hardware configurations and camera setups for each domain. These descriptions are concatenated with the task instruction and processed by the Florence-Base encoder, enabling the model to explicitly attend to embodiment-specific variations. Table 12 lists the language prompt templates used across domains.

| Domain | Language Prompts |
|---|---|
| RoboMind-Franka | Embodiment: Single Franka, Camera Setup: Top View, Freq: 30Hz |
| RoboMind-UR | Embodiment: Single UR, Camera Setup: Top View, Freq: 30Hz |
| Droid-Left | Embodiment: Single Franka, Camera Setup: Left View / Wrist View, Freq: 15Hz |
| Droid-Right | Embodiment: Single Franka, Camera Setup: Right View / Wrist View, Freq: 15Hz |
| AGIBOT | Embodiment: AGIBOT, Camera Setup: Head View / Wrist View, Freq: 30Hz |
| RoboMind-Agilex | Embodiment: AgileX, Camera Setup: Head View / Wrist View, Freq: 30Hz |
| RoboMind-Dual-Franka | Embodiment: Dual Franka, Camera Setup: Front View / Wrist View, Freq: 30Hz |

Table 12: Language prompts designed to provide embodiment- and camera-specific descriptions for each domain in the preliminary experiments.

## J EVALUATION DETAILS IN REAL-WORLD EXPERIMENTS

In this section, we provide detailed descriptions of our real-world evaluation setups. We adapt *X-VLA-0.9B* to three distinct robotic embodiments, each selected to validate different aspects of the model's adaptability:

**WidowX** for pick-and-place experiments. *X-VLA-0.9B* finetuned on BridgeData is directly deployed to evaluate its ability to perform robust manipulation on a compact platform. We conduct comprehensive evaluations to assess both manipulation performance and language-instruction following in real-world settings, as illustrated in Fig 13 and each task is evaluated for 10 times.

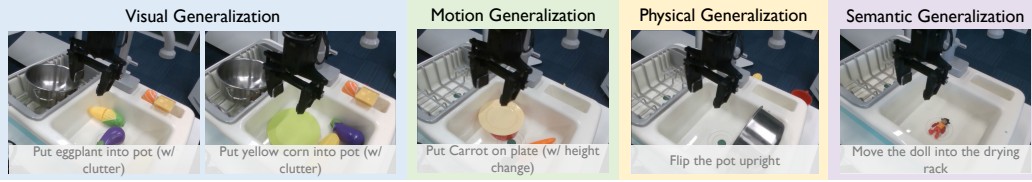

Figure 13: Illustration of tasks used in the WidowX pick-and-place experiments. The selected tasks evaluate different aspects of generalization—Visual, Motion, Physical, and Semantic—following the setup in Open-VLA (Kim et al.).

**AgileX** for dexterous manipulation tasks. As discussed in Appendix F, this setup is designed to test dexterous, fine-grained control on a bi-manual platform equipped with wrist-mounted cameras.

**AIRBOT** for parameter-efficient finetuning experiments. AIRBOT is unseen during pretraining. We specifically collect only 200 demonstrations for a cloth-picking task, making it a challenging low-

resource setting. This experiment highlights the adaptability of our two-step adaptation procedure under strict data and resource constraints.

Fig. 14 shows the hardware setups for these experiments. Each embodiment is equipped with distinct camera setups, enabling us to construct a heterogeneous deployment environment for validation.

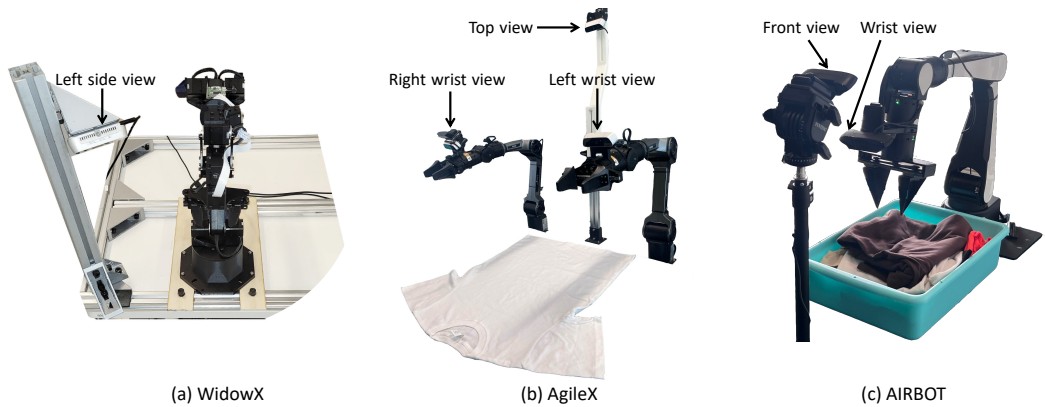

Figure 14: Illustration of the hardware setups used in real-world experiments. We evaluate on three robotic embodiments, including WidowX, Agilex, and AIRBOT, covering diverse camera configurations and task domains to form a heterogeneous validation environment.

## K    TRAINING DETAILS OF BASELINES IN REAL-WORLD EXPERIMENTS

In this section, we provide the training details of the real-world baselines.

$\pi_0$ **in cloth-folding task** is finetuned from the official base $\pi_0$ model using the Soft-Fold dataset described in Appendix F. The model is trained with a total batch size of 32 across 4 A100 GPUs, requiring approximately 60 hours to complete 150,000 gradient steps.

$\pi_0$ **in PEFT experiments** is finetuned from the official $\pi_0$ base model using the official LoRA configuration. We apply LoRA with rank 16 and $\alpha = 16$ to both the attention and FFN modules within the PaliGemma-2B VLM. For the action expert, we use rank 32 and $\alpha = 32$. Training is performed with a total batch size of 32 across 4 A800 GPUs, taking approximately 7 hours to complete 30,000 gradient steps.

**ACT in cloth-folding task** is trained from scratch using the Soft-Fold dataset described in Appendix F. The model is trained with a total batch size of 256 on 8A100 GPUs. Since the model capacity of ACT is not high as large model such as X-VLA and $\pi_0$, we train ACT approximately 1M gradient steps for better training.

## L    EVALUATION DETAILS ON AUTONOMOUS DRIVING SIMULATION BENCHMARK

We evaluate our method on the large-scale real-world autonomous driving benchmark NAVSIM (Dauner et al., 2024) using closed-loop assessment. Following the official evaluation protocol, we report the PDM score (higher indicates better performance), which aggregates five key metrics: *NC* (no-collision rate), *DAC* (drivable area compliance), *TTC* (time-to-collision safety), *Comfort* (acceleration/jerk constraints), and *EP* (ego progress). All methods are tested under the official closed-loop simulator, and results are averaged over the public test split. As an end-to-end VLA model, our method achieves superior performance over specialized methods designed for autonomous driving, with detailed scores reported in Table 13.

## M    EVALUATION DETAILS ON ROBOTICS SIMULATION

We report detailed scores for each simulation benchmark in Table 14-18.

| NAVSIM | | | | | | |
|---|---|---|---|---|---|---|
| Methods | NC | DAC | EP | TTC | C | PDMS |
| Transfuser (Chitta et al., 2022) | 97.7 | 92.8 | 79.2 | 92.8 | **100.0** | 84.0 |
| UniAD (Hu et al., 2023) | **97.8** | 91.9 | 78.8 | **92.9** | **100.0** | 83.4 |
| UniVLA (Wang et al., 2025a) | 96.9 | 91.1 | 76.8 | 91.7 | 96.7 | 81.7 |
| X-VLA (Ours) | 97.5 | **96.5** | **82.2** | **92.9** | **100.0** | **87.3** |

Table 13: Detailed results on NAVSIM benchmark.

| Simpler | | | | | | | | | | | | | | |
|---|---|---|---|---|---|---|---|---|---|---|---|---|---|---|
| Visual Matching (Google Robot) | | | | | Visual Aggregation (Google Robot) | | | | | Visual Matching (WidowX Robot) | | | | |
| Coke | Near | Open | Put | Average | Coke | Near | Open | Put | Average | Spoon | Carrot | Blocks | Eggplant | Average |
| 98.3 | 97.1 | 69.5 | 56.5 | 80.4 | 85.5 | 79.8 | 61.9 | 75.7 | 75.7 | 100 | 91.7 | 95.8 | 95.8 | 95.8 |

Table 14: Detailed results on Simpler benchmark.

| Libero | | |
|---|---|---|
| | Libero-Spatial | 98.2 |
| | Libero-Object | 98.6 |
| | Libero-Goal | 97.8 |
| | Libero-Long | 97.6 |
| | Average | 98.1 |

Table 15: Details on Libero.

| Calvin (ABC→D) | | |
|---|---|---|
| | 1 | 97.1 |
| | 2 | 92.6 |
| | 3 | 88.5 |
| | 4 | 84.4 |
| | 5 | 78.8 |
| | Average | 4.43 |

Table 16: Details on Calvin.

| VLABench | | |
|---|---|---|
| | In Distribution | 67.8 |
| | Cross Category | 25.1 |
| | Common Sense | 48.2 |
| | Semantic Instruction | 63.1 |
| | Average | 51.1 |

Table 17: Details on VLABench.

| RoboTwin-2.0 | | | | | | | | |
|---|---|---|---|---|---|---|---|---|
| Task | Easy | Hard | Task | Easy | Hard | Task | Easy | Hard |
| Adjust Bottle | 97.0 | 56.0 | Open Microwave | 85.0 | 57.0 | Place Object Stand | 78.0 | 33.0 |
| Beat Block Hammer | 78.0 | 18.0 | Pick Diverse Bottles | 27.0 | 25.0 | Place Phone Stand | 80.0 | 9.00 |
| Blocks Ranking RGB | 79.0 | 26.0 | Pick Dual Bottles | 30.0 | 27.0 | Place Shoe | 70.0 | 51.0 |
| Blocks Ranking Size | 42.0 | 9.00 | Place A2B Left | 62.0 | 21.0 | Press Stapler | 70.0 | 13.0 |
| Click Alarmclock | 96.0 | 69.0 | Place A2B Right | 54.0 | 17.0 | Put Bottles Dustbin | 0.00 | 1.00 |
| Click Bell | 100 | 61.0 | Place Bread Basket | 75.0 | 39.0 | Put Object Cabinet | 78.0 | 82.0 |
| Dump Bin Bigbin | 94.0 | 59.0 | Place Bread Skillet | 82.0 | 17.0 | Rotate QRcode | 78.0 | 52.0 |
| Grab Roller | 99.0 | 66.0 | Place Burger Fries | 98.0 | 47.0 | Scan Object | 60.0 | 44.0 |
| Handover Block | 27.0 | 30.0 | Place Can Basket | 58.0 | 18.0 | Shake Horizontally | 99.0 | 100.0 |
| Handover Mic | 100 | 38.0 | Place Cans Plasticbox | 100 | 85.0 | Shake Bottle | 99.0 | 99.0 |
| Hanging Mug | 34.0 | 15.0 | Place Container Plate | 98.0 | 60.0 | Stack Blocks Three | 22.0 | 15.0 |
| Lift Pot | 99.0 | 75.0 | Place Dual Shoes | 98.0 | 28.0 | Stack Blocks Two | 87.0 | 55.0 |
| Move Can Pot | 50.0 | 44.0 | Place Empty Cup | 98.0 | 34.0 | Stack Bowls Three | 80.0 | 42.0 |
| Move Pillbottle Pad | 52.0 | 29.0 | Place Fan | 72.0 | 27.0 | Stack Bowls Two | 83.0 | 10.0 |
| Move Playingcard Away | 94.0 | 57.0 | Place Mouse Pad | 19.0 | 3.00 | Stamp Seal | 52.0 | 13.0 |
| Move Stapler Pad | 58.0 | 35.0 | Place Object Basket | 50.0 | 0.00 | Turn Switch | 40.0 | 13.0 |
| Open Laptop | 85.0 | 73.0 | Place Object Scale | 39.0 | 13.0 | Average | 70.0 | 39.0 |

Table 18: Detailed results on RoboTwin-2.0 benchmark.

