# OpenReview forum: "X-VLA: Soft-Prompted Transformer as Scalable Cross-Embodiment Vision-Language-Action Model"
_ICLR.cc/2026/Conference — ICLR 2026 Poster_

### Official Review · Reviewer_F3Do · 2025-10-31

**Soundness:** 4
**Presentation:** 3
**Contribution:** 3
**Rating:** 8
**Confidence:** 5

**Summary:**

This paper addresses the significant challenge of training a single, generalist Vision-Language-Action (VLA) model on large-scale, heterogeneous, and cross-embodiment robotic datasets. The authors propose **X-VLA**, a flow-matching-based VLA architecture. The core contribution is a **"Soft Prompt" mechanism**, which assigns a unique set of learnable embeddings to each distinct data source (i.e., hardware configuration). These prompts condition a standard Transformer backbone, enabling it to handle the heterogeneity arising from different embodiments while learning a generalist policy. The model is trained using a two-phase pretraining and domain adaptation strategy. The authors demonstrate that their 0.9B parameter model, X-VLA-0.9B, achieves new state-of-the-art (SOTA) performance across a comprehensive suite of six simulation benchmarks and three real-world robot platforms , and shows remarkable parameter-efficient finetuning (PEFT) capabilities.

**Strengths:**

1.  **Clear Problem, Elegant Solution:** The paper tackles a critical and timely problem in large-scale robotics: how to effectively leverage diverse, heterogeneous datasets. The proposed "Soft Prompt" mechanism  is an elegant and simple solution that integrates cleanly into a standard Transformer architecture. This approach is more deeply integrated than using domain-specific action heads (Fig 2a)  and more flexible and scalable than relying on hand-crafted language prompts (Fig 2c).

2.  **Exceptional Empirical Results:** This is the paper's most significant strength. The evaluation is both comprehensive and highly convincing.
    * The model achieves SOTA results on an impressively broad set of **6 simulation benchmarks** (Libero, Simpler, VLABench, RoboTwin-2.0, Calvin, and NAVSIM). The performance gains shown in Table 2 are substantial (e.g., 95.8% on Simpler-WidowX vs. 71.9% prior SOTA; 87.3% on NAVSIM vs. 81.7% prior SOTA).
    * The work is validated on **3 distinct real-world robotic platforms** (WidowX, Agilex, AIRBOT), demonstrating strong real-world applicability. This includes impressive performance on a challenging dexterous cloth-folding task.
    * The introduction and planned release of the new **"Soft-Fold" dataset** for dexterous manipulation is a valuable contribution to the community.

3.  **Strong Adaptation and Parameter Efficiency:** The parameter-efficient finetuning (PEFT) results are a key highlight. The ability to achieve performance comparable to a fully finetuned 3B parameter model ($\pi_{0}$) while tuning only 9M parameters (1% of the 0.9B model) is a powerful demonstration of the model's capabilities. This strongly supports the claim that the backbone learns a truly "embodiment-agnostic" generalist policy.

4.  **Thorough Analysis and Ablations:** The paper does an excellent job of justifying its design choices.
    * Figure 4 provides a direct comparison against other methods for handling heterogeneity, showing that soft prompts lead to more stable training and lower validation error.
    * Table 1 offers a clear ablation path, demonstrating the positive contribution of each key component (e.g., soft-prompt, scaling up, two-step adaptation).
    * The scaling plots in Figure 5 are promising, showing consistent improvement with model size, data diversity, and data volume, with no sign of saturation.
    * The T-SNE visualization in Figure 8 provides compelling qualitative evidence that the soft prompts are successfully capturing meaningful, embodiment-specific information.

**Weaknesses:**

1.  **Limited Methodological Novelty of Soft Prompts:** The core mechanism, "Soft Prompting," is not a new invention. It is a well-established technique from the NLP community (e.g., Lester et al., 2021)  and has been used in multi-task and multi-domain learning , as the paper acknowledges. The novelty here lies in its *application* to the *cross-embodiment robotics* domain. While the *results* are novel and SOTA, the *methodological* leap is more of an effective and well-executed application rather than a new fundamental mechanism.

2.  **Clarity on Soft Prompt Implementation:** Key details about the soft prompt's architecture are not fully clear in the main text. Figure 1 shows them feeding into the Transformer block, but the exact mechanism (e.g., concatenation, addition) is ambiguous. This is clarified in Appendix C / Figure 10, which shows the "Soft Prompt" tokens are concatenated with the multi-modal tokens and control tokens. Moving this crucial implementation detail from the appendix to the main methodology section would significantly improve the paper's clarity and self-containedness.

3.  **Ambiguity in T-SNE Visualization:** The T-SNE plot in Figure 8  is a key piece of qualitative evidence, but it's not entirely clear what is being plotted. The legend lists 7 data sources , but there are many dots for each source. Given that Figure 5 (center) explores "Prompt Length", it's plausible that the prompt is a *sequence* of tokens. If so, does each dot in the T-SNE plot represent a single token from that sequence? Clarifying this is important for correct interpretation.

**Questions:**

1.  The soft prompt is described as "a set of learnable embeddings" , and Figure 5 (center) explores "Prompt Length". Does this confirm that each "soft prompt" is a *sequence* of learnable tokens (e.g., a matrix of size $L \times D_{model}$)? If so, what prompt length was used for the final X-VLA-0.9B model?

2.  Regarding the T-SNE visualization in Figure 8: What do the individual dots represent? If the prompt is a sequence of tokens, does each dot represent one token from that prompt sequence, for its corresponding data source?

3.  The two-step adaptation procedure  involves (1) prompt warm-up and (2) joint policy adaptation. The PEFT experiments (Table 3)  use LoRA. Is this LoRA applied *in addition* to the two-step prompt tuning? As a baseline, did the authors try *only* tuning the new soft prompt (while keeping the entire 0.9B backbone frozen), which would be a more standard "prompt tuning" PEFT approach?

4.  In the architecture (Section 4.1) , the paper mentions disentangling streams, with a pretrained VLM for the main view/language and a "shared vision backbone" for auxiliary views. Is this "shared vision backbone" the *same* vision encoder from the VLM, or is it a separate vision encoder?

---

> ### Author Response · Authors · 2025-11-19
> **Rebuttal to Reviewer F3Do**
>
> Thank you for the valuable feedback and for recognizing the motivation, scalability, and performance of our framework. Below, we provide detailed responses addressing each of your comments:
>
> > W1: Limited Methodological Novelty of Soft Prompts...
>
> Please kindly refer to `General Response 1`.
>
> > W2: Clarity on Soft Prompt Implementation: Key details ...
>
> Thank you for the valuable suggestion. We agree that the soft-prompt implementation details are important for clarity, and we have moved this description into the main methodology section (Sec 4.3) in the revised version of the paper.
>
> >W3: Ambiguity in T-SNE Visualization: The T-SNE plot in Figure 8 is ...
>
> Yes, you are right:
>
> 1. The soft prompt for each domain is implemented as a sequence of learnable tokens.
> 2. Each dot in the T-SNE plot represents a single token from the sequence of a specific soft prompt.
>
> Since each data source includes multiple tokens to encode embodiment heterogeneity, jointly visualizing all tokens across embodiments intuitively reflects both intra and inter-embodiment relationships. We have strengthened the description of this part in **Sec 5.3** of the revised paper. Thanks for your valuable feedback.
>
>
> > Q1: The soft prompt is described as "a set of learnable embeddings" ...
>
> The prompt length used in the final X-VLA-0.9B model is 32, determined through systematic scaling experiments conducted across different model sizes and prompt lengths (see Figure 6 in revised paper). We have added further clarification of these implementation details, including the prompt length and other hyperparameters of X-VLA-0.9B, in **Sec. 4.3** of the revised paper.
>
>
> > Q2: Regarding the T-SNE visualization in Figure 8: What...
>
> please kindly refer to the answer for W3.
>
> > Q3: The two-step adaptation procedure involves (1) prompt warm-....
>
> Yes, you are right — LoRA is applied **in addition** to the two-step prompt tuning procedure. We also evaluated a more standard *“prompt tuning only”* baseline, where a newly initialized soft prompt is tuned while the entire 0.9B backbone remains frozen. However, this approach yielded very poor performance on Simpler-WidowX (below 10%). This is expected, as tuning only the soft prompt results in fewer than 50K trainable parameters, which is insufficient for aligning the action space and task distribution in complex manipulation settings.
>
> To further support this point, we conducted an additional experiment in which we still apply *“prompt tuning only”* but unfreeze the last layer of the backbone to increase the learnable capacity. The results are shown below:
>
>
> | Tunable Parameters      | Prompt Only (32K) | Prompt + Linear Head (~70K) | Prompt + LoRA (9M) | Prompt + Unfreeze Last Layer (25M) |
> | ----------------------- | ----------------- | --------------------------- | ------------------ | ---------------------------------- |
> | Success Rate on Simpler | 0%                | 8.3%                        | 54.2%              | **68.9% (SOTA)**                   |
>
>
> We have incorporated these additional results into **Appendix C.5** of the revised paper.
>
>
> > Q4: In the architecture (Section 4.1) , the paper mentions ...
>
> The shared vision backbone is the same vision encoder from the VLM.

---

### Official Review · Reviewer_R55g · 2025-11-01

**Soundness:** 4
**Presentation:** 4
**Contribution:** 4
**Rating:** 8
**Confidence:** 4

**Summary:**

The paper proposes X-VLA, a vision-language-action model that leverages "soft prompts" (learnable embeddings of heterogeneous hardware platforms) to effectively address heterogeneity across different robot platforms when learning multi-task policies. The method learns domain-specific soft prompt embeddings that encode individual embodiments, with a shared VLM backbone that utilizes the soft prompts to ultimately generate high-quality actions across multiple embodiments and tasks. The model is pretrained on 290K demonstrations from seven robot platforms that include both single-arm and dual-arm manipulation setups. Experimental evaluations show strong, state-of-the-art performance across the benchmarks, which include LIBERO, SIMPLER, CALVIN, and others.

**Strengths:**

* The paper conducts extensive and thorough experimental evaluation which includes six simulation benchmarks and three real-world robotic platforms. A variety of prior methods are compared against, including $\pi_0$, OpenVLA, Octo, ACT, and several other VLAs (UniVLA, GR00T-N1, FLOWER, UniVLA, etc.).
* The results across all settings are consistently strong, nearly outperforming all other prior methods and even matching $\pi_0$ (3B parameters) with a PEFT version that use only 9M trainable parameters.
* The experimental analysis on various components of X-VLA is comprehensive, with various ablations and a study on scaling behavior across model and dataset size. Further, the qualitative and quantitative discussions on the effects of the learned soft prompts are informative and support the claims that the author makes regarding the ability to encode heterogeneous morphologies in a shared structured latent space.
* The paper also introduces the Soft-Fold bimanual cloth folding dataset with 1,200 demonstrations.
* The uncut video on the project website showcasing two hours of t-shirt folding demonstrates impressive, robust performance of X-VLA across an extended period of time.
* I appreciate the transparent discussion of failed attempts to address heterogeneity across mixed data sources presented in Appendix E. Reporting unsuccessful attempts may provide useful information to readers.

**Weaknesses:**

* Soft prompts, the core contribution of the paper, is borrowed from prior natural language processing research (Lester et al., 2021), which suggests limited technical novelty. While the application of soft prompts to vision-language-action models in robotics is novel, the paper reads more like an engineering project that obtained a strong model through various careful techniques employed during training.
* Some important details are missing: How many tokens are used in the soft prompts, how is the number chosen, and how does performance vary with respect to the number of tokens used in the soft prompt? In addition, how is the temporal downsampling rate for pretraining chosen, and how does that affect policy performance? Further, what effect does the prompt warm-up phase discussed in Section 4.2.1 have on final performance? The paper would be strengthened by additional discussions addressing these questions.

**Questions:**

* How does X-VLA perform under OOD generalization settings?
* When does X-VLA fail during deployment, and what do the failure modes look like qualitatively?
* Can a soft prompt be used for zero-shot deployment on an unseen robot if the soft prompt was trained for a seen robot that is very similar to the new robot?

---

> ### Author Response · Authors · 2025-11-19
> **Rebuttal to Reviewer R55g**
>
> Thanks for the valuable feedback and for recognizing our rigorous experimental setup, strong performance, and contribution toward democratizing embodied AI research. Below, we provide detailed responses addressing each of your comments:
>
> > W1: Soft prompts, the core contribution of the paper, ...
>
> Please refer to the `General Response 1`.
>
> > W2: Some important details are missing: ....
>
> **1. How many tokens are used in the soft prompts, how is the number chosen, and how does performance vary with respect to the number of tokens used in the soft prompt?**
>
> In X-VLA-0.9B, we use 32 soft-prompt tokens by default. This choice is supported by systematic scaling experiments conducted across different model sizes, with the corresponding results reported in Fig. 6. We have also added further discussion of the implementation details of X-VLA-0.9B in **Sec 4.3** of the revised paper. Thanks for your valueble feedback.
>
>
> **2. how is the temporal downsampling rate for pretraining chosen, and how does that affect policy performance?**
>
> The temporal downsampling rate(or the size of prediction window) for pretraining is selected through a careful grid-search using a small-scale preliminary setup. Specifically, we pretrained a 0.3B version of X-VLA on our curated data recipe for 100K iterations and evaluated the resulting model using PEFT on Simpler-WidowX (as validation losses under different prediction targets are not directly comparable). The results are summarized below:
>
> | Size of Prediction Window | 1s | 2s | 4s | 8s |
> |-----|----------------|-------------|----------------|--------------|
> | **Success rate on Simpler** |      0%      |     8.3%      |       29.16%       |        27.08%      |
>
> The results and further discussion have been incoperated into the revised version of our paper in **Appendix C.4**.
>
>
> **3. what effect does the prompt warm-up phase discussed in Section 4.2.1 have on final performance?**
>
> The effect of the prompt warm-up phase is presented and discussed in Table 1. We provide the corresponding results below, and kindly refer the reviewers to that table for detailed comparisons and further discussion.
>
> | methods | Acc on Simpler-WidowX |
> |-----|----------------|
> |w/o warm-up phase|   89.6% |
> | **w warm-up phase** |      **95.8%(+6.2%)**      |
>
> > Q1: How does X-VLA perform under OOD generalization settings?
>
> The OOD generalization performance of X-VLA is demonstrated across multiple simulation benchmarks. Results on Simpler, RobotWin-Hard, and CALVIN indicate strong robustness to OOD visual scenario shifts as well as variations in language instructions. For real-world evaluation, we deployed X-VLA-0.9B—tuned only on the Soft-Fold dataset—in a fully OOD setting characterized by high dynamic range and substantial background disturbances, without any additional finetuning. X-VLA exhibited strong robustness to these interferences and maintained reliable generalization (please refer to **the updated demo video** available on the website linked in the abstract). For further discussion on OOD and embodiment generalization, please refer to Appendix N.
>
>
> >Q2: When does X-VLA fail during deployment, and what do the failure modes look like qualitatively?
>
> The failure modes of X-VLA observed during real-world deployment can be summarized as follows:
> 1. The current X-VLA-0.9B model is constrained by its relatively modest parameter count and learning capacity. Consequently, it may exhibit limited reasoning ability when handling long-horizon tasks or highly complex language instructions.
>
> 2. Similar to many VLA-based policies, X-VLA may struggle with precise positioning and may occasionally fail to execute accurate grasps. Nevertheless, we observe a consistent recovery behavior: the policy often retries its motions and can eventually “escape” failure states. This behavior is particularly evident in our cloth-folding video, where the model gradually corrects its actions over repeated attempts.
>
> We provide additional discussion of the limitations of X-VLA in Appendix N, and we kindly refer the reviewer to that section for more details.
>
>
> >Q3: Can a soft prompt be used for zero-shot deployment ...
>
> It depends. Currently, we only use 7 data sources in our pretraining setup, which is relatively limited to bring absolute zero-shot generalization. However, we observe positive transferability by leveraging pretrained soft-prompts, as shown in Figure 9 in our paper and `General Response 2`. Therefore, as the diversity of pretraining data increases, the transferability of soft prompts can also be substantially improved.

---

### Official Review · Reviewer_PCNL · 2025-11-01

**Soundness:** 3
**Presentation:** 3
**Contribution:** 2
**Rating:** 6
**Confidence:** 3

**Summary:**

The paper proposes X-VLA policy for learning across different robot setups using soft prompts for each data source. The pipeline is trained with a flow-matching objective for action chunk generation. The training involves first pretraining on ~290k episodes across seven setups, then prompt warm-up with frozen backbone for new embodiment and finally joint PEFT (LoRA) that tunes ~1% of parameters. The evaluation shows strong performance across six simulation benchmarks and three real-robot platforms. The learned soft prompts cluster based on hardware in t-SNE plot.

**Strengths:**

The idea of soft prompts is motivated well, and is shown to be better than the language templates and HPT-style projections.
The overall proposal seems a scalable solution, as it is based on standard transformer with shared params for encoders and validation errors suggest room to grow.
The paper explores how reduced LR leads to better performance for their experiments and report consistent gains in performance across the simulated benchmarks.

**Weaknesses:**

Prompts are queried by dataset/hardware ID; this assumes reliable domain labels during training and may risk overfitting to source IDs unless carefully regularized. There are no experimental evidence of zero-shot generalization to unseen tasks with known robots and unseen robots.

While the experiment results has a wide coverage, the current focus is on tabletop tasks with focus on the end-effector movements, and do not require much reasoning of the differences in the embodiment. It is unclear if the proprioceptive states outputs are actually needed for the evaluation tasks.

The results use PEFT with 1% tunable params that is promising for on-robot finetuning, but this has been compared on quite saturated Libero benchmark. The results reported have no error bars or confidence intervals.

**Questions:**

What does the prefix "high-dimensional" observation stream and "low-dimensional" proprioceptive–action stream mean? Can you clarify what is it in comparison to, like does it have more frames for history or multi-perspective viewing? How big are the "low-dimensional" proprioceptive–action vectors, and how do they differ with different embodiments?

Are the results statistically significant?

How useful are the learned soft prompts? While the paper provides a t-SNE plot to show how embeddings cluster by hardware, it is unclear how important they are for reusing in a new instance of a task. For example, if a soft prompt embedding is used with another task same embodiment or another embodiment, what is the performance degradation on the task?

---

> ### Author Response · Authors · 2025-11-19
> **Rebuttal to Reviewer PCNL-1/2**
>
> We appreciate the reviewer for the valuable feedback and for recognizing the motivation, scalability, and performance of our framework. Below, we provide detailed responses addressing each of your comments:
> > W1: Prompts are queried by dataset/hardware ID...
> 1. **Reliable domain IDs:**
> The pre-training and post-training data used for X-VLA, in particular, is curated from recent, high-quality, open-sourced datasets [1-6] or directly derived from benchmarks, which have been widely examined by the community. This greatly reduces the risk of noisy or inconsistent domain labeling.
> 2. **Risk overfitting to source IDs:**
> Our experiments extensively validate the generalization capability of X-VLA with learned soft prompts. Specifically:
>   - Unseen tasks with known robots:
> We refer the reviewer to our simulation benchmark experiments—covering CALVIN, Simpler, and VLABench—as well as our real-robot evaluations on WidowX. Specifically, the CALVIN ABC-D benchmark evaluates generalization from environments A/B/C to an entirely unseen environment D; Simpler assesses robustness to variations in visual scenes through real-to-sim evaluation; and VLABench measures generalization to unseen tasks with novel language instructions~[4, 5, 6]. In addition, our real-world evaluation is explicitly designed to validate generalization from multiple perspectives, as illustrated in Fig. 13.
>   - Unseen robots:
> Please refer to `General Response 2`. There, we evaluate pretrained soft prompts on new-domain adaptation tasks, demonstrating that pretrained soft prompts can indeed be transferred across embodiments, with performance strongly dependent on the similarity of hardware configurations.
> More importantly, we also include real-world experiments on an unseen robot, AIRBOT, and validate the generalization ability of X-VLA on this platform under the PEFT setting; please refer to **Figure 8** for details.
>
> > W2: While the experiment results has a wide coverage, ...
>
> - We'd like to clarify that reasoning about differences in embodiments is essential when **pretraining on mixed datasets that exhibit strong hardware heterogeneity**. Even though many tasks focus on predicting EE movements, different robotic platforms correspond to distinct world coordinate systems and physical meanings. Addressing these heterogeneity using a learning-based aproach is exactly the main focus and contribution of our framework. In Figure 9, we provide clear evidences that the learned soft prompts captures semantically meaningful embodiment configurations, which actually need reasoning about the embodiment-specific information through raw robotic data. Our results in Table 1 further demonstrate the significant performance gains during pretraining brought by our proposed method.
> - Furthermore, once an embodiment-agnostic policy is learned, **adapting to a novel downstream task also requires reasoning about the new hardware configuration as well as the proprioceptive states**. This is supported by the results in `General Response 2`, that explicit embodiment reasoning via soft prompt learning is crucial for effective policy adaptation.
>
> > W3: The results use PEFT with 1% tunable params that is promising..
>
> We also conducted PEFT experiments on the Simpler Benchmark [5], which is a more recent and challenging manipulation benchmark, and we have reported these results in Table 3. Our findings show that X-VLA with only 9M param tuned achieves performance comparable to Pi0 on Simpler as well.
> Nevertheless, we acknowledge that the PEFT experiments could be strengthened with more comprehensive evaluations to mitigate variances during training. To address this concern, we extended the PEFT experiments to five independent runs, each using different training and evaluation seeds. The aggregated results are shown below, which are consistent with our originally reported scores, and the performance variations are also reasonably small. These additional results have been incorporated into **Table 3** of the revised paper
>
> |  | Libero-Spatial | Libero-Goal | Libero-Object | Libero-Long |
> |-----|----------------|-------------|----------------|--------------|
> | **Avg** |      95.8  |       95.2    |         96.3      |     83.7        |
> | **Std** |      0.4   |        0.8     |          0.3      |       0.5       |
>
> [1] Droid: A large-scale in-the-wild robot manipulation dataset. arXiv:2403.12945, 2024
>
> [2] Robomind: Benchmark on multi-embodiment intelligence normative data for robot manipulation. arxiv.org/abs/2412.13877.
>
> [3] Agibot world colosseo: A large-scale manipulation platform for scalable and intelligent embodied systems. arXiv:2503.06669, 2025
>
> [4] Calvin: A benchmark for language-conditioned policy learning for long-horizon robot manipulation tasks.RAL, 2022
>
> [5]  Evaluating real-world robot manipulation policies in simulation. CoRL, 2025.
>
> [6]  Vlabench: A large-scale benchmark for language-conditioned robotics manipulation with long-horizon reasoning tasks. ICCV-2025

---

> ### Author Response · Authors · 2025-11-19
> **Rebuttal to Reviewer PCNL-2/2**
>
> > Q1: What does the prefix "high-dimensional" observation...
>
> - The “high-dimensional” observation stream refers to the encoded image and language tokens produced by a pretrained VLM. In contrast, the “low-dimensional” proprioceptive–action stream includes proprioceptive information (e.g., current end-effector or joint states), noised actions, and time embeddings used for flow matching. These two streams differ substantially in both semantic content and dimensionality: the hidden size of the high-dimensional stream is 1024, whereas the dimensionality of the proprioceptive–action stream typically ranges from 10 to 20, depending on the embodiment.
> - Different embodiments introduce significant discrepancies in both streams. For the high-dimensional stream, as mentioned by the reviewer, different robots may provide different numbers of camera views as well as camera position and angles. For the low-dimensional stream, the differences are even more pronounced: each embodiment has its own unique action space and proprioceptive state representation. As a result, the dimensionality and physical meaning of the low-dimensional vectors vary significantly across robots. Please refer to Fig 10 in our paper for more details about the architecture as well as the overall pipeline.
>
>
>
> > Q2: Are the results statistically significant?
>
> **Yes, all reported results are statistically significant.**
> For all simulation-based evaluations, we strictly follow the official evaluation protocols of each benchmark. These protocols require a large number of rollouts to ensure statistical reliability—for example: 1,000 rollouts for Calvin, 2,000 for LIBERO, and 5,000 for RobotWin. For real-world deployment, we also conduct extensive and rigorous evaluations. Each task includes at least 10 rollouts, and the dexterous cloth-folding experiment involves over 2 hours of continuous rollouts.
>
> We believe that our paper provide **one of the most comprehensive validation protocols in the embodied AI literature to date**. Additional evaluation details and full experimental setups can be found in Appendix J.
>
> > Q3: How useful are the learned soft prompts?....
>
> 1. As discussed above, the task generalization capability of the learned soft prompt is well supported by our evaluations on the CALVIN, Simpler, and VLABench simulation benchmarks, as well as the real-world experiments on WidowX. Together, these settings cover a broad range of manipulation tasks and variations.
> 2. Generalization across embodiments is further demonstrated by the results reported in `General Response 2`, where the learned soft prompts exhibit clear interpretability as well as strong transferability across different robot embodiments.

---

### Author Response · Authors · 2025-11-19
**General Response**

We sincerely thank all reviewers for their time, thoughtful feedback, and engagement throughout the review process.

**We have submitted a revised version of the paper that fully incorporates the reviewers’ feedback and includes all additional experiments conducted during the rebuttal. In summary:**

1. **Expanded empirical validation and discussion:**
- Extended PEFT experiments on Libero with five independent seeds in Table 3 (Reviewer PCNL),
- Expanded experiments on transferibility of pretrained soft-prompt in Figure 10 (Reviewer PCNL/R55g),
- New ablations on prediction window selection in Appendix C.4 (Reviewer R55g),
- New evaluations of PEFT variants (prompt-only, LoRA, unfreezing layers) in Appendix C.5 (Reviewer F3Do)

2. **Improved methodology clarity:**
- We further clarified the design and implementation of soft prompts, including prompt length selection(Figure 6), T-SNE visualizations(Sec 5.3) and full hyperparameter details for X-VLA-0.9B(Figure 5 and Sec 4.3).
3. **Additional real-world validation:**
- We expanded discussion and evidence for real-robot OOD generalization, including newly updated demo videos and clearer explanation of failure modes and recovery behaviors on the website linked in the abstract.

Below, we provide further clarification regarding our core contributions and the design of the soft-prompt mechanism.


### 1. Core Contribution and Technical Novelty of Soft Prompts

We agree that soft prompts are originated from the NLP community. However, our core contribution goes beyond simply introducing an existing technique into robotic learning or limiting the work to an engineering effort. Our aim is to highlight and address **the fundamental challenge of pretraining on highly heterogeneous robotic datasets**—a setting that has received limited attention but is increasingly critical for scaling embodied AI.

Rather than naively applying soft prompts, our work introduces nontrivial insights into architectural design choices, including a clean and scalable Transformer-based decoder and an elaborately structured novel input-encoding pipeline that is fully compatible with soft prompts. Moreover, we provide extensive empirical evidence showing that embodiment-aware soft prompting, together with other components of our framework, such as the pretraining and adaptation strategy, enables large-scale VLA pretraining across diverse robot embodiments. Please refer to Section 3 and Appendix D.2 for additional details.

We believe this direction offers a practical and extensible path for future methods to scale on heterogeneous datasets—an unavoidable challenge as the field moves toward more general-purpose robot learning.


### 2. The importance and transferability of Soft-Prompt.

The effectiveness of the proposed soft-prompt mechanism extends beyond improving pretraining stability. Our experiments show that soft prompts encode embodiment-aware information that can be reused and transferred across different robot embodiments.

To further validate this, we extended the experiments presented in Fig. 9 of the original paper during the rebuttal, where we evaluate the success rate of X-VLA-0.9B finetuned for Simpler-WidowX using PEFT with different soft-prompt configurations. To intuitively illustrate the importance of meaningful soft prompts—and the transferability of pretrained prompts—we consider four settings:
(1) a randomly initialized soft prompt,
(2) a soft prompt learned on the dual-arm Agibot robot,
(3) a soft prompt learned on the single-arm UR5 robot, and
(4) an adapted WidowX soft prompt.

We report the result from finetuning for both 50K and 200K  on the demonstrations. The formal performance reflects the fast adaptability and transferability enabled by the soft prompt while the latter corresponds to the performance achieved after sufficient and fully converged finetuning.
The results show that adapted soft prompts not only yield substantial performance gains, but also exhibit a degree of transferability that correlates with hardware similarity—providing direct evidence that the learned prompts capture meaningful embodiment structure. Please refer to **Fig. 10** in the revised paper for further details and discussion.

|  Used Soft Prompt  |  Random initialized  | Agibot(Dual arm) | UR-5(Single Arm) | Adapted WidowX(Single Arm) |
|-----|----------------|-------------|----------------|--------------|
| **Best Acc after 50K iters tuning** |      0%     |     26.0%      |    39.6%   |     **48.75%**  |
| **Best Acc after 200K iters tuning** |     43.8%       |  44.8%  |    48.75%      | **54.2%**|

---

### Author Response · Authors · 2025-11-26
**A Kindly Reminder: Looking Forward to Further Discussion**

Dear Reviewers,
Thank you once again for your time and effort throughout the review process. We have submitted the rebuttal and the revised paper in response to your concerns, which have been available for some time now. Please let us know if there are any further issues or questions.
Best regards

---

### Meta-Review · Area_Chair_bAFF · 2026-01-07

**Summary:**

The reviewers praised the paper's strong empirical contributions, including new SOTA results across 6 simulation benchmarks and 3 real-world platforms, impressive PEFT efficiency (matching a 3B model with only 9M tuned parameters), comprehensive ablations, and a new bimanual cloth-folding dataset with long-horizon real-world demos. Most concerns were effectively addressed in the rebuttal, significantly strengthening clarity, experiments, and generalization evidence. While technical novelty remains limited (soft prompts adapted from NLP), the non-trivial application to heterogeneous robotics data, scalable architecture, and outstanding empirical performance outweigh this concern, justifying acceptance.

**Reviewer Concerns:**

Most concerns were effectively addressed in the rebuttal:

- Clarity issues (prompt implementation, t-SNE details, stream definitions, temporal downsampling, warm-up phase effects) were clarified or moved to the main paper with new figures/tables.
- Experimental gaps were filled with added results: multi-seed error bars on PEFT (Table 3), extended transfer experiments (Fig. 10), prompt-only/LoRA ablations (Appendix C.5), and detailed OOD/failure mode discussions (Appendix N + updated videos).
- Generalization to unseen tasks/embodiments was strengthened via references to existing benchmarks (CALVIN, Simpler, VLABench) and new transfer evidence.

Outstanding: The novelty concern remains partially unaddressed—while the rebuttal emphasizes non-trivial application to heterogeneous robotics data and architectural integration, the core soft-prompt mechanism is still an adaptation of prior NLP work rather than a fundamental invention.

**Reviewer Scores:**

- Reviewer PCNL (original: 6): Likely increases to 7–8; rebuttal directly resolved their main concerns on generalization, statistical significance, error bars, and prompt utility/transfer.
- Reviewer R55g (original: 8): Likely remains 8 or increases to 9; rebuttal provided requested details on prompt length, downsampling, warm-up, OOD performance, and failure modes.
- Reviewer F3Do (original: 8): Likely remains 8–9; rebuttal clarified implementation details, t-SNE, and added PEFT ablations.

---

### Decision · Program_Chairs · 2026-01-26

Accept (Poster)